# The PI3K-Akt-mTOR and Associated Signaling Pathways as Molecular Drivers of Immune-Mediated Inflammatory Skin Diseases: Update on Therapeutic Strategy Using Natural and Synthetic Compounds

**DOI:** 10.3390/cells12121671

**Published:** 2023-06-20

**Authors:** Tithi Roy, Samuel T. Boateng, Mohammad B. Uddin, Sergette Banang-Mbeumi, Rajesh K. Yadav, Chelsea R. Bock, Joy T. Folahan, Xavier Siwe-Noundou, Anthony L. Walker, Judy A. King, Claudia Buerger, Shile Huang, Jean Christopher Chamcheu

**Affiliations:** 1School of Basic Pharmaceutical and Toxicological Sciences, College of Pharmacy, University of Louisiana at Monroe, Monroe, LA 71209, USA; royt@warhawks.ulm.edu (T.R.); boatengst@warhawks.ulm.edu (S.T.B.); sbchamcheu@gmail.com (S.B.-M.); yadavrk@warhawks.ulm.edu (R.K.Y.); bockcr@warhawks.ulm.edu (C.R.B.); folahanjt@warhawks.ulm.edu (J.T.F.); awalker@ulm.edu (A.L.W.); 2Department of Toxicology and Cancer Biology, Center for Research on Environmental Diseases, College of Medicine, University of Kentucky, Lexington, KY 40536, USA; burhan1129@gmail.com; 3Division for Research and Innovation, POHOFI Inc., Madison, WI 53744, USA; 4School of Nursing and Allied Health Sciences, Louisiana Delta Community College, Monroe, LA 71203, USA; 5Department of Pharmaceutical Sciences, School of Pharmacy, Sefako Makgatho Health Sciences University, P.O. Box 218, Pretoria 0208, South Africa; xavier.siwenoundou@smu.ac.za; 6Department of Pathology and Translational Pathobiology, LSU Health Shreveport, 1501 Kings Highway, Shreveport, LA 71103, USA; judy.king@belmont.edu; 7College of Medicine, Belmont University, 900 Belmont Boulevard, Nashville, TN 37212, USA; 8Department of Dermatology, Venerology and Allergology, Clinic of the Goethe University, 60590 Frankfurt am Main, Germany; claudia.buerger@kgu.de; 9Department of Biochemistry and Molecular Biology, Louisiana State University Health Sciences Center, 1501 Kings Highway, Shreveport, LA 71130, USA; shile.huang@lsuhs.edu; 10Department of Hematology and Oncology, Louisiana State University Health Sciences Center, 1501 Kings Highway, Shreveport, LA 71130, USA; 11Feist-Weiller Cancer Center, Louisiana State University Health Sciences Center, Shreveport, LA 71130, USA

**Keywords:** acne, PI3K-Akt-mTOR signaling pathway, skin inflammation, phytochemicals, psoriasis, atopic dermatitis, wound healing, biologics and targeted therapy, antioxidants, flavonoids and bioactive natural products/nutraceuticals

## Abstract

The dysregulated phosphatidylinositol-3-kinase (PI3K)-Akt-mammalian target of rapamycin (mTOR) signaling pathway has been implicated in various immune-mediated inflammatory and hyperproliferative dermatoses such as acne, atopic dermatitis, alopecia, psoriasis, wounds, and vitiligo, and is associated with poor treatment outcomes. Improved comprehension of the consequences of the dysregulated PI3K/Akt/mTOR pathway in patients with inflammatory dermatoses has resulted in the development of novel therapeutic approaches. Nonetheless, more studies are necessary to validate the regulatory role of this pathway and to create more effective preventive and treatment methods for a wide range of inflammatory skin diseases. Several studies have revealed that certain natural products and synthetic compounds can obstruct the expression/activity of PI3K/Akt/mTOR, underscoring their potential in managing common and persistent skin inflammatory disorders. This review summarizes recent advances in understanding the role of the activated PI3K/Akt/mTOR pathway and associated components in immune-mediated inflammatory dermatoses and discusses the potential of bioactive natural products, synthetic scaffolds, and biologic agents in their prevention and treatment. However, further research is necessary to validate the regulatory role of this pathway and develop more effective therapies for inflammatory skin disorders.

## 1. Introduction to the Structure and Function of the Skin

The skin is the largest and outermost organ in humans, located at the boundary between the interior and exterior environment, and plays a crucial role in the body’s functions. It is composed of distinct tissue layers, including the epidermis, dermis, and hypodermis. The epidermis consists of four to five cell layers primarily composed of keratinocytes, which produce keratins essential for maintaining skin integrity. Other cell types in the epidermis include melanocytes, Langerhans, mastocytes, and Merkel cells (Figure 1A) [1,2,3,4,5,6,7,8,9,10,11,12,13,14,15,16].

The epidermis is composed of distinct layers of keratinocytes at different stages of maturation and function, including the germinative and proliferative cell layer (stratum basale; SB), the spinous or prickle cell layer (stratum spinosum; SS), the granular cell layer (stratum granulosum; SG), and the cornified or horny cell layer (stratum corneum, SC). In palmoplantar skin, the stratum lucidum comes before the horny layer to maintain healthy skin with functional barriers (Figure 1B) [3,5,17,18,19,20,21,22].

The dermis supports the epidermis and contains blood and lymphatic vessels, sensory elements, and various cells. Skin adnexal structures, such as sweat glands, sebaceous glands, and hair follicles, arise from invaginations of the epidermis. Stem cells can be found in the bulge region of hair follicles, the basal layer of the interfollicular epidermis, and in sebaceous glands. The dermis is a dense connective tissue underlying the epidermis that provides skin’s strength and flexibility through its extracellular matrix mainly composed of collagen, elastin, fibrillin, and glycoproteins [5]. It is composed of the papillary and reticular layers and contains blood vessels, hair follicles, nerve endings, sweat, and sebaceous glands. Additionally, it houses diverse skin resident cell types, including fibroblasts responsible for the production of collagen, elastin, and GAGs for tissue elasticity and histiocytes, such as macrophages, lymphocytes, and mast cells that provide immune responses [5].

The hypodermis is the innermost skin layer, mainly comprised of adipose tissue and other structures, providing insulation and energy storage. The three skin tissue compartments interact via various mechanisms to maintain healthy skin and a functional barrier [5,21,22].

The skin is located on the exterior of the body and serves various functions, including protecting against external insults, such as chemical and biological agents, physical factors, and injuries, as well as regulating the exchanges of body fluids and substances involved in homeostasis, metabolism, and energy storage [2,21,23,24,25,26]. The skin’s visible and exterior location is also important in social and sexual relationships.

Dysregulation of the skin’s tightly regulated homeostasis and immune responses due to modifiable or non-modifiable risk factors can lead to various chronic inflammatory dermatoses.

## 2. Risk Factors Related to the Pathogenesis of Inflammatory Skin Disorders

Cutaneous diseases, including inflammatory dermatosis, are influenced by both modifiable (environmental) and non-modifiable (genetic) risk factors, which can disrupt skin homeostasis [5]. Environmental factors including humidity, temperature, air pollutants, and exposure to UV radiation—the foremost environmental risk factor—cause a diverse range of adverse effects on the skin such as hyperplasia, inflammation, premature aging, pigmentation disorders, and chronic inflammatory skin conditions. Cutaneous UV-mediated adverse inflammatory processes are triggered via the activation of skin cells, inducing immune mediators and pro-inflammatory cytokines [27,28]. Other environmental factors, such as microbial infection, oxidative stress, and injuries, also impair skin homeostasis. Genetic risk factors play a significant role in the development of inflammatory and autoimmune skin diseases, with mutations in genes encoding skin structural proteins found to be associated with some skin conditions [27,28,29,30,31,32,33,34]. Beyond genetics, epigenetic mechanisms such as DNA methylation, histone modifications, and non-coding RNA molecules that regulate gene activity, can also influence cutaneous cell differentiation, inflammation, and other processes relevant to skin disorders [35] (PMID: 32787761). Moreover, DNA hypomethylation, site-specific DNA hypermethylation, and micro-RNA (miRNA) expression have been reported both in psoriasis and skin cancer. Importantly, histone modifications, including acetylation, methylation, and phosphorylation, have also been associated with skin inflammation and disease pathogenesis. Developing and delivering natural and synthetic products and biologics targeting dysfunctional targets are prominent therapeutic strategies to mitigate these diseases. 

Skin homeostasis is maintained by signaling pathways such as PI3K/Akt/mTOR, which regulate multiple physiological processes. Dysregulated activation of this pathway has been observed to be associated with the pathophysiology of cutaneous cancers and immune-mediated dermatoses. Several natural phytochemicals, synthetic scaffolds, and biologic molecules have the potential to modulate these pathways and prevent/treat major immune-mediated skin diseases [5,34].

UV radiation can have negative effects on the skin, but other applications such as phototherapy or combining with conventional therapies like psoralen can provide benefits for treating chronic cutaneous inflammatory disorders like acne, hidradenitis suppurativa (HS), atopic dermatitis (AD), lichen planus (LP), psoriasis, morphea, scleroderma, and vitiligo [36,37,38]. In addition to UV radiation, other environmental factors like microbialatopic infections, oxidative stress, and injuries can also cause skin problems [5,39].

Research has shown that genetic risk factors play significant roles in the development of inflammatory and autoimmune skin diseases, particularly psoriasis and AD, which are non-modifiable factors [40,41]. Studies involving families, populations, and epidemiological research have shown that certain genetic risk patterns of skin diseases like psoriasis are linked to human leukocyte antigens (HLAs) after conducting candidate gene and genome-wide linkage analyses within and outside of the major histocompatibility complex (MHC) region [41,42]. Furthermore, mutations in several genes that encode skin structural proteins have been associated with skin conditions such as AD [43,44,45,46].

Recent advances in our understanding of the genetic and molecular mechanisms underlying most cutaneous inflammatory disorders have led to the development of natural and synthetic products, as well as biologics, which target dysfunctional pathways to mitigate disease and enhance skin homeostasis, ultimately reinforcing skin integrity [47,48]. Skin homeostasis is maintained through the regulation of skin stem cells located in the dermis, epidermis, hair follicles, and other areas, which are replenished to replace damaged or aging cells [49,50,51]. Various signaling pathways, including Hedgehog (HH), Notch, MNK, MAPK, transforming growth factor β (TGF-β), Wnt, PI3K/Akt/mTOR, Syk, among others, regulate these stem cell pools [52,53,54,55]. The PI3K/Akt/mTOR axis plays a critical role in regulating multiple physiological processes, such as cell growth, proliferation, autophagy, apoptosis, migration, invasion, differentiation, metabolism, and angiogenesis [56,57,58,59]. Dysregulation of this signaling pathway has been associated with the pathophysiology of cutaneous cancers [5], as well as immune-mediated dermatoses such as acne, psoriasis, vitiligo, and scleroderma [60,61]. In the following sections, we summarize the relevance of the dysregulated PI3K-Akt-mTOR and associated signaling pathways to certain immune-mediated skin disorders. We also discuss the mechanisms of action of several natural phytochemicals, synthetic scaffolds, and biologic molecules in modulating these pathways and their potential in preventing and treating major immune-mediated skin diseases. 

## 3. Overview of the Structure and Function of the PI3K-Akt-mTOR Signaling Pathway

The PI3K-Akt-mTOR signaling pathway plays a critical role in various physiological and pathological conditions, particularly in cell growth, differentiation, and survival. The inter-connectivity of PI3K, Akt, and mTOR with other signaling cascades, such as HIFs, AP-1, JNK, Notch, FOXO, RAS, ERK1/2, Wnt, and Rho small GTPases, has been widely recognized [5,61]. This pathway serves as a hub in cellular response to external stimuli like EGF, FGF, insulin, and IGF-1. It is also connected with upstream regulators such as chemokine receptor-9, TLRs, and IL-6 and plays crucial roles in cell growth, proliferation, survival, migration, and differentiation. Therefore, it has emerged as a central regulator of tissue and organ homeostasis and a key driver of immune-mediated dermatoses [62].

In the 1980s, the importance of PI3K was discovered when a physio-functional relationship was observed between its activity and the transformation activity of the viral oncogene and with activated protein-tyrosine kinases [63]. PI3Ks are characterized as a family of lipid kinases capable of phosphorylating the inositol ring 3′-OH group in inositol phospholipids. The PI3K consists of three distinct classes of regulatory, functional, and structural molecules, with class I PI3K being the most extensively studied [62,63,64,65].

### 3.1. Class I PI3Ks and the Key Intracellular Effectors

Class I PI3Ks are the most studied PI3K family members, with similar structures and substrate specificities, including PI, PIPs, and PIP2. Class I PI3Ks are a family of lipid kinases that play important roles in cellular physiology. They are composed of a catalytic domain (p110 subunit) and a regulatory subunit (p85) and are divided into two subfamilies, IA and IB. They are activated via cell surface receptors and phosphorylate PIP2 to generate PIP3, which regulates several cellular processes and activates downstream signaling pathways, including Akt/mTOR. The different isoforms of p110 have distinct functions in skin cells and are expressed in various cell types involved in inflammatory dermatoses [63,66,67,68,69,70,71,72,73,74,75,76,77,78,79,80,81,82].

The p110δ isoform of PI3K is expressed in cell types such as keratinocytes, synovial fibroblasts, and endothelial cells, which are frequently involved in the pathophysiology of inflammatory dermatoses [81,82]. PI3K activation occurs via cell surface receptors that bring them into proximity with their substrates (PI, PIPs, and PIP2) [44]. Activation of class IA PI3K via RTKs is mediated by the p85 regulatory subunit, which directly binds to cell membrane receptors, stimulating the receptors and leading to the recruitment of the p85 subunit and subsequent activation of PI3K p110 by binding to RTKs’ intracellular phosphorylated tyrosine residues [81,82]. Once activated, PI3K p110 phosphorylates PIP2 to generate PIP3, which regulates various cell physiological processes and downstream Akt/mTOR signaling pathways by binding and phosphorylating/activating other proteins, including Akt [5,83].

### 3.2. Class II and Class III PI3Ks

Class II PI3Ks contain obligatory regulatory subunits encompassing a Ras-binding motif and have substrates such as PIP2s and phosphatidylinositol (PI). They are able to interact with other possible adaptor proteins and may be activated via cytokine receptors, integrins, and RTKs. The Class II PI3Ks serve a role in the cellular cortex by modulating cell migration, membrane trafficking, glucose transport, insulin signaling, as well as receptor internalization [84,85].

Furthermore, the only known class III PI3K is Vps34, which is thought to regulate membrane trafficking functions such as autophagy flux, endosome-lysosome maturation, endosomal protein sorting, and cytokinesis using PI as a substrate. The downstream targets of class III PI3Ks/Vps34 are also controlled in response to cellular stress, growth and survival, and accessibility of amino acids [84,85].

### 3.3. PKB/Akt

PKB (Protein kinase B)/Akt, a serine/threonine (Ser/Thr) kinase, regulates many cellular and physiological functions including protein synthesis, glucose metabolism, cell cycle progression, cell proliferation, and survival (Reviewed in [5] and Figure 2).

Indeed, Akt plays an important role in maintaining epidermal homeostasis and sustaining keratinocyte function, promoting keratinocyte differentiation and survival, and is fully functional upon activation at two specific amino acids via phosphorylation at Ser^473^ and Thr^308^. The process is initiated upon binding of PIP3 to Akt, leading to its recruitment to the plasma membrane, where it gets phosphorylated in Thr^308^ by phosphoinositide-dependent kinase-1 (PDK1) [5]. From there, mTORC2 (see below and Figure 2) phosphorylates Akt at Ser^473^ resulting in fully activated Akt, which subsequently phosphorylates mTORC1 [5]. Akt regulates a plethora of signaling proteins and cellular processes. For example, Akt also phosphorylates and inhibits pro-apoptotic proteins (BAD and procapase-9) and FOXO transcription factor. FOXO functions as a negative modulator for cell proliferation; inhibition of FOXO increases cell proliferation. Furthermore, Akt indirectly induces several anti-apoptotic genes by affecting NF-κB [6]. Akt also negatively regulates the tuberous sclerosis complex (TSC) activity [62]. The TSC1/2 complex acts as a GTPase-activating protein (GAP) for Rheb (Ras homolog enriched in the brain) small GTPase [5] GTP-bound Rheb directly binds to and activates mTORC1 [5]. Inactivation of TSC1/2 increases the level of GTP-Rheb, thus activating mTORC1 [5]. Akt also inhibits another mTOR inhibitor called the proline-rich Akt substrate of 40Kda (PRAS40) by dissolving the bond between mTOR and PRAS40 [5] (See Figure 2).

### 3.4. The Mammalian Target of Rapamycin (mTOR)

mTOR is a Ser/Thr kinase present in mammalian cells, and it plays a vital role in regulating cell growth, survival, and proliferation by controlling the biosynthesis of proteins, nucleotides, and lipids [5,84]. It is a part of the PI3K/Akt pathway and exists as two multi-protein complexes, mTORC1 and mTORC2, with different compositions and functions. mTORC1 regulates mRNA translation by phosphorylating S6K1, S6K2, and 4E-BP, while mTORC2 controls cell growth and survival through Akt phosphorylation [86]. Positive growth regulators, such as growth factors and their receptors, including EGF/EGFR, IGF-1/IGF-1R, and VEGF/VEGFR, activate mTOR signaling via the PI3K/Akt pathway, while negative regulators, including PTEN, TSC1/2, AMPK, HIF1, and REDD1, inhibit mTOR signaling. TSC2 is phosphorylated by Akt, which leads to its release of Rheb and activates mTOR. In addition, mTORC1 and mTORC2 have been shown to regulate skin morphogenesis, epidermal barrier formation, and other pathways such as Wnt and keratin-17 [5,87,88] (see Figure 2).

## 4. Role of the PI3K-Akt-mTOR and Related Networks in Skin Development and Homeostasis

Accordingly, alterations in normal homeostatic regulated pathways can modulate protein synthesis to impact skin cell growth and proliferation, resulting in phenotypically diverse skin diseases including immune-mediated inflammatory dermatoses [88,89,90,91]. Understanding the molecular mechanisms by which extracellular cues are integrated into the cells to modulate cell physiology is vital for the diagnosis of diseases and the design of novel therapeutic regimens for inflammatory dermatoses. Amongst the diverse signaling pathways, the PI3K/Akt/mTOR signaling pathways [92,93] (Figure 2) have emerged as clinically important therapeutic targets for chronic inflammatory dermatoses. This centrally located and closely interconnected pathway pivotally regulates many essential cellular physiologic functions, including cell metabolism, nucleotide, lipid and protein synthesis, cell growth, proliferation, autophagy, apoptosis, migration, and angiogenesis [94,95]. These signaling molecules have recently been linked with diverse human inflammatory diseases with chronic cutaneous manifestations as highlighted below and represent important molecular targets for therapeutic agents’ discovery.

### 4.1. Diagnostic and Therapeutic Consequences of the PI3K-Akt-mTOR and Related Cascades as Drivers of Inflammatory Dermatoses

Owing to the key role played by the PI3K-Akt-mTOR pathway in cellular physiology, tissue homeostasis, and pathologic modification, is implicated in the pathogenesis of a spectrum of acquired and heritable human diseases with cutaneous manifestations. It can be activated by either overexpression/amplification or mutations of diverse genes encoding the associated proteins in the pathway. Some of these gene products, including Akt, PI3K p110α, PTEN, and inositol polyphosphate 4-phosphatase type II (INPP4B), have been reported in cancers including skin cancer (reviewed in [5,63]). The PI3K-Akt-mTOR pathway is an emerging and potential therapeutic target in cutaneous inflammatory disorders but, so far, few drugs have been tested and approved for their treatments, mainly due to adverse toxicity effects. Some publications have discussed the effects of natural products and/or synthetic scaffolds on the PI3K-Akt-mTOR pathway components with regard to hyperproliferative or specific inflammatory skin disorders [63] or skin cancer type [5]. Equally, they have been reported to be modulated and overexpressed in inflammatory skin conditions, several of which are characterized by cutaneous tissue hyperplasia and inflammation, including skin conditions such as acne vulgaris, alopecia, AD, buruli ulcer, hidradenitis suppurativa, oral/lichen planus, psoriasis, systemic sclerosis, morphea, scleroderma, vitiligo, hypertrophic scar, wound healing, etc. [96,97,98,99] and [84,100,101]. It is promising to develop and deliver therapeutic agents targeting the dysfunctional signaling molecules toward reinforcing skin integrity, enhancing homeostasis, and ultimately treating diseases [47,48]. Therefore, there is a dire need to understand the in-depth pathognomonic mechanism and to explore newer specific mechanism-based therapeutic agents against various immune-mediated skin disorders, which remain key global issues [5]. Recent technological innovations in animal and in vitro disease models, high throughput drug screening, and several novel promising approaches are summarized in the sections below. Here, we provide a comprehensive review of the emerging role of the PI3K-Akt-mTOR and associated signaling pathways as a therapeutic target in immune-mediated dermatoses and highlight the potential of bioactive natural, synthetic, and other products targeting the networks for chemoprevention and/or chemotherapy.

#### 4.1.1. Role of the PI3K-Akt-mTOR and Allied Networks, and Their Therapeutic Targeting in Major Inflammatory Skin Diseases

##### Atopic Dermatitis

Atopic dermatitis (AD) or atopic eczema is a common chronic and often relapsing inflammatory skin disease characterized by pruritic/itchy and red eczematous skin lesions [102]. The AD skin becomes sensitized to allergens and irritants, which results from complex pathological factors including genetic predisposition, defects in the skin barrier protein filaggrin [103,104,105,106], microbial infection, autoimmunity, as well as environmental factors [107].

In AD pathogenesis, Th1 and Th2 cell responses play a role, with Th2 cells activating B lymphocytes and mast cells and Th1 cells maintaining the chronic phase [108]. Cytokines from both types of cells impact filaggrin expression in cutaneous keratinocytes, with Th2 cytokines suppressing its expression [102,109]. Foxp3+ regulatory T cells suppress disease progression, and signaling pathways downstream of FcεRI, such as MAPK/ERK and PI3K-Akt-mTOR, are involved. mTOR signaling negatively regulates Foxp3 expression, with mTORC1 being responsible for Th1 differentiation and mTORC2 regulating Th2 activation via Akt [109,110,111,112,113].

The Stat5-PI3K-Akt signaling pathway effector molecules, including mTOR and S6K, act as potential therapeutic targets for diseases involving basophils, mast cells, or neoplastic mast cells. Akt1 is important for normal cornified envelope formation and reduced Akt1 activity observed in AD skin is associated with decreased expression of proteases and filaggrin, as well as impaired skin barrier function. This decrease in Akt1 activity has been linked to the development of erythematous and scaly AD skin lesions. Some reports suggest that targeting this pathway may have a beneficial effect on AD pathogenesis [60,114,115,116,117,118].

##### Therapeutic Targeting the PI3K/Akt/mTOR for Treating Atopic Dermatitis (AD)

Atopic dermatitis (AD) currently lacks a definitive cure, with patient management mostly focused on symptom relief [119]. Efforts are being made to discover and develop mechanistic drugs that target attractive molecular targets, with dysregulation of the PI3K-Akt-mTOR pathway being a major contributor to the disease. A plethora of interventions including small molecules, natural products, biologics, and phototherapy approaches targeting this pathway have proven useful in managing AD. For instance, synthetic compounds have been used to modulate or induce skin barrier-related structural proteins such as filaggrin, a novel therapeutic target for AD, with JTC801 and rapamycin being some of the compounds used to induce filaggrin expression. Otsuka et al. showed that treatment of human immortalized and normal keratinocytes with JTC801, a derivative of 4-aminoquinoline resulted in the induction of filaggrin expression [119]. Yang et al. reported that topically applied ointment of rapamycin, an mTORC1 inhibitor, suppressed immune responses in dermatophagoides farina body extract-induced AD in NC/Nga mice [120]. Natural products such as *Actinidia arguta* and Baicalein have also shown promise in alleviating AD symptoms by targeting the mTOR signaling pathway [115,121]. Red ginseng extracts (RGE) have demonstrated anti-inflammatory and anti-allergic effects, with ginsenosides Rg3 and Rh1 being used therapeutically to decrease IgE-induced phosphorylation of S6K. Osada-Oka et al. examined the anti-inflammatory effects of RGE on IgE or IFN-γ and reported an anti-allergic effect of RGE, showing significant inhibition of 2,4-dinitrofluorobenzene (DNFB)-induced ear swelling and scratching behavior [101]. Activation of the high-affinity IgE receptor (FCεRI) in KU812 human basophils (known initiators of IgE-mediated chronic allergic inflammation) and induction of mTOR/S6K phosphorylation were studied by treatment with RGE. Of interest, the ginsenosides Rg3 and Rh1 therapeutically decreased IgE-induced phosphorylation of S6K (p-S6K), despite no effect of phosphorylation of mTOR [102,122]. Fisetin, a naturally occurring bioactive flavonol, has also shown potential in diverse AD models. Taken together, targeting the PI3K-Akt-mTOR pathway and associated pathways via interventions such as Actinidia argute, RGE, rapamycin, 4-aminoquinoline, dupilumab, and phototherapy are some of the available options that have been explored in the management of AD.

##### Psoriasis

Psoriasis is a common chronic, recurrent, and currently incurable, autoimmune-mediated skin disease with a dynamic, complex, and incompletely understood pathophysiology. Although the etiology of psoriasis remains to be elucidated, it is known to be caused by abnormal proliferation of epidermal keratinocytes, which contributes to the immune imbalance, and infiltration of activated immune cells leading to chronic skin lesions characterized by clearly distinguishable raised erythematous plaques covered with white scales. Psoriasis affects approximately 2–4% of the population worldwide with more than 8 million (roughly 3% of adults) in the United States alone [123,124,125]. While not fatal, psoriasis often shows significant comorbidities and a substantial decrease in patients’ quality of life, with about 25% increased risk for patients to progress to debilitating diseases such as psoriatic arthritis [126] Even though psoriasis is mostly reported in adult age, it has been increasingly reported in early age with a prevalence of 0.5% and 2% in children and adolescent, respectively [127], with the report of gradual disease progression in the entire course of life [128]. The severity of psoriasis is dependent on the rate and extent of epidermal keratinocyte hyperplasia, infiltration of several different leukocytes (e.g., monocytes, dendritic cells, and T lymphocytes) in both the epidermis and dermis, resulting in dilation of blood vessels and neo-angiogenesis in psoriatic patients [98,128,129]).

The multifactorial nature of psoriasis results from a combination of genetic predisposition and environmental triggers, and treatments with single agents or single target therapies have limited benefits to patients. So, it is of great importance to identify novel disease targets and develop effective targeted therapies for psoriasis. Several recent studies have highlighted that among the many signaling networks implicated in psoriasis, deregulation of the PI3K/Akt/mTOR and associated signaling pathways critically contribute to disease initiation and progression (reviewed in [98]). Deregulated PI3K/Akt/mTOR signaling has been shown to impact both the structural skin barrier components as well as inflammatory disease components of psoriasis [101,130]. In psoriasis, uncontrolled keratinocyte proliferation is associated with activation of the mTOR pathway and involves different Th-1, Th-17, and Th-22 immune cells, and their effector cytokines (e.g., IL-1β, IL-6, IL-17A, TNF-α) that act on keratinocytes [100]. The involvement of Th-17 cells and their secreted cytokines, such as IL-17A, IL-22, and IL-23, etc., is well documented [59,83,126,129,131,132]. Components of the PI3K/Akt/mTOR pathway and its downstream effector molecules (e.g., S6K1) have been reported to be hyperactivated in stained psoriasis patient skin lesions compared to healthy control or non-lesional skin. Activated signalinal molecules showed differential localization within the epidermis, and were thought to contribute to an imbalance in the Th1/Th2/Th17 axis associated with psoriasis pathogenesis [99,100,114]. mTOR is activated by different cytokines including IL-1β, and IL-23 that initiate the growth of Th17 cells whose products including IL-17A, IL-17F, and IL-22 mediate inflammatory effects [132]. More recently, we identified differential roles of various IL-17 isoforms, showing that IL-17E (IL-25) activated mTOR while IL-17A did not [133]. Keratinocyte-released IL-36 also plays an important role by upregulating the PI3K-Akt-mTOR pathway [60]. T-cells derived from patients with psoriasis have been reported to have amplified receptors and increased activities for the PI3K/Akt/mTOR pathway in psoriatic skin lesions [60,84,132]. Comparatively, in normal healthy skin, the mTOR pathway is shunted at the initiation of the epidermal keratinocyte differentiation process, whereas in psoriatic epithelium, the keratinocyte differentiation process is not halted due to sustained activation of the PI3K/Akt/mTOR pathway. Upon activation of the mTOR network, diverse cytokines and other pro-inflammatory mediators, such as IL-6, CXCL8, and IL-22, are released, which increases the inflammation and proliferation of cells commonly seen in psoriasis [60,84,134,135]. For instance, IL-22 induces increased keratinocyte proliferation, while elevated chemokines including CXCL8, CXCL10, and CCL20 have been found in inflamed areas of psoriatic skin lesions. While CXCL8 and CXCL10 synergize to attract neutrophils and T-cells to inflamed psoriatic skin lesions, CCL20 activates Type 17 T-cells (Th-17) and assists in the production of IL-17 and IL-22 [134,135]. More important to the kit, downstream effectors of mTORC2 such as the epidermal Rho small GTPases (Ras-related C3 botulinum toxin substrate 1–3 (Rac1–3), particularly hyperactivation of epidermal RAC1 has been shown to promote psoriasiform inflammation that closely resembles human psoriasis [136].

##### Targeting the PI3K/Akt/mTOR for Treating Psoriasis

There is no final cure for psoriasis, so it is necessary to identify novel targets and effective treatments that can alleviate symptoms having minimal side effects. To date, mTOR inhibitors have been extensively studied for the treatment of inflammatory skin diseases as well as cancers. Rapamycin/sirolimus, a classic mTOR inhibitor, inhibits mTORC1 specifically and allosterically, which forms a complex with the intracellular receptor FK506 binding protein 12 (FKBP12) and then binds the FKBP12-rapamycin binding (FRB) domain of mTOR, disrupting the assembly of mTORC1 [5]. Rapamycin is far from the perfect mTOR inhibitor as it inhibits mTORC1 only partially and activates Akt through a feedback loop in certain cases, leading to some resistance occurring and incomplete therapy. Rapamycin was shown to block the phosphorylation of S6K1 and 4E-BP1, essential for the synthesis of cell growth proteins and to effectively inhibit tuberous sclerosis in which mutations of *TSC1* and *TSC2* genes cause constitutive activation of mTOR [5]. Additionally, rapamycin was also found to decrease the level of VEGF and thus the migration of dendritic cells from the skin towards lymph nodes, with the cells being particularly responsible for mTOR activation via Toll-like receptor7 (TLR7) activation [108]. Despite these promising effects, resistance to rapamycin in psoriatic patients has been reported because of the partial inhibition of mTORC1 and the activation of Akt through a feedback loop. When in the presence of mTOR inhibitors, intact, healthy skin shows little response as proliferation has already occurred, but when mTOR inhibitors are in contact with unhealthy skin or disrupted skin, the wounds are disrupted, and the healing process is slowed down or halted in severe cases [60,84,87]. In a recent study, the mTOR inhibitor rapamycin has been shown to inhibit cell proliferation in culture and ameliorate symptoms in imiquimod-induced psoriatic animal models [137]. We demonstrated that topical application of rapamycin decreased the activation of the PI3K-mTOR cascade and normalized the expression of epidermal differentiation markers in imiquimod-induced psoriasiform dermatitis model in mice, suggesting the utility of rapamycin as a topical remedy for psoriasis [101,108]. Another mTOR inhibitor everolimus, a rapamycin analog, has also shown potential in treating psoriasis [138] alone or in combination with tacrolimus in a patient with severe refractory psoriasis [139].

However, rapamycin did trigger the research and investigation of novel alternative mTOR inhibitors [84,140]. Vitamin D has also been used traditionally to treat mild-moderate forms of psoriasis, which interferes with psoriasis pathophysiology by targeting several pathways, for instance, by inhibiting the activation of inflammatory responses by the plasmacytoid dendritic cells (pDC) pathway. The mechanism is related to the fact that decreased presence of cutaneous pDCs in psoriasis results in reduced production of pro-inflammatory mediators produced and subsequent declined inflammation. An analog of vitamin D 1α,25-dihydroxyvitamin D_3_-3-bromoacteate has been shown to inactivate Akt and eventually mTOR, resulting in decreased production of IL-22 and psoriasis-like characteristics [141].

More recently, small molecule oral preparations targeting intracellular signaling that may prove effective have been developed. Our studies of in vitro and in vivo preclinical psoriasis mouse models demonstrated that the natural dietary antioxidant delphidine (see section below) effectively inhibits both mTORC1 and mTORC2 via inhibition of PI3K and the S6K1/IRS-1 feedback loop. Other signaling pathways identified to be deregulated in psoriasis, such as STAT1/3, AMPK, Ras/MAPK, and NF-κB, also directly or indirectly activate mTOR [100]. The activation of AMPK, an important regulator of cellular energy homeostasis and metabolism, was reported to potently inhibit cell growth and proliferation via mTOR inhibition and to induce cell death by activating the p53/p21 pathway [142].

A naturally occurring flavonoid luteolin showed an inhibitory effect against proinflammatory mediators. Its structural analog, tetramethoxyluteolin, was found to have a more potent effect than luteolin [143]. It has been shown that stimulation of human keratinocytes by TNF resulted in mTORC1-mediated secretion of proinflammatory mediators IL-6, CXCL8, and VEGF, and tetramethoxyluteolin could block the mTOR activation and inhibit the secretion of inflammatory mediators [143]. Because of its beneficial effect against psoriasis, tetramethoxyluteolin-containing lotion has become commercially available for the treatment of mild psoriasis [143].

Delphinidin has been reported to have beneficial properties in diverse human diseases including inflammatory skin disorders such as psoriasis [144,145,146,147]. Using biophysical, in vitro cell-free, cell culture, and in vivo murine models, we have demonstrated the antipsoriatic effect of delphinidin and identified that its therapeutic mechanism also includes acting as a dual PI3K/Akt and mTOR inhibitor [140,144,145,146,147]. In fact, delphinidin binds to mTORC1 in the same binding area as rapamycin but does not need to bind to FKBP12 in order to carry out its effect. Furthermore, delphinidin can inhibit PI3K, resulting in reduced release of IL-22 and alleviating psoriasiform symptoms [140]. In addition, Kwon et al. showed that delphinidin treatment of mouse skin suppressed UVB-induced overexpression of PI3K/MAPKK4 as well as expression of COX-2 [147]. Mitra et al. reported that IL-22-induced proliferation of NHEK and fibroblast-like synoviacytes (FLS) is dependent on the PI3K-Akt-mTOR signaling pathway, suggesting the possibility of developing novel treatments targeting this pathway in chronic immune-mediated diseases such as AD, psoriasis, and rheumatoid arthritis. As a proof of concept, these authors utilized NVP-BEZ235 (a dual PI3K/mTOR kinase inhibitor) and rapamycin to ably inhibit both IL-22-induced proliferation and phosphorylation of Akt and mTOR in both NHEK and FLS [148].

Activation of AMPK signaling by metformin, a first-line anti-diabetic drug, was found to inhibit mTOR and proliferation of human keratinocytes. Using HaCaT cells, it was shown that metformin inhibits the expression of inflammatory cytokines (IL-6 and TNF-α) and downregulates the growth factor VEGF thus inhibiting the inflammatory response in psoriasis by inhibiting mTOR [131].

Other natural products and extracts have also been investigated to treat psoriasis. It has been described that fisetin possesses a therapeutic effect against in vitro psoriasis-like changes and chronic photo-damage in vivo, by suppressing the expression of MMP-1, MMP-2, and COX-2, and restoring the barricading characteristic of the skin [149]. Baicalin was found to mitigate the common symptoms of psoriasis, such as epidermal thickening, erythema, and desquamation induced by IL-17A, IL-22, and IL-23 [150].

The Chinese licorice strain *Glycyrrhiza glabara* has been investigated by utilizing the glycyrrhizic acid metabolite 18-ß glycyrrhetinic acid, which possesses antioxidative, antitumor, and anti-inflammatory properties. The metabolite was shown to directly inhibit mTOR, resulting in an increase in regulatory T-cells within the spleen, which decreased cutaneous psoriasis-like features and promoted healthy skin care [151].

Total glycosides of paeony (TGP), derived from the dried root of the Chinese medicine *Paeonia lactifloria Pallas*, contain the active ingredient paeoniflorin, which was found to inhibit the PI3K/Akt/mTOR signaling, reducing the function and activity of the immune cells and exhibiting anti-inflammatory effects [152].

Matrine is derived from the traditional Chinese medicine *Sophora flavescens* Aiton. Sophora has been traditionally used to treat a variety of ailments such as hepatitis B, tumors, inflammation, eczema and other skin disorders. Matrine was reported to inactivate the PI3K/Akt/mTOR pathway, mitigating psoriasis symptoms [153].

Erianin is a natural bibenzyl compound derived from *Dendrobium chrysotuxum*. Erianin has been shown to possess anti-tumor and anti-psoriasis activity partly by suppressing the Akt/mTOR pathway. The exact mechanism through which erianin acts to inhibit the Akt/mTOR signaling is still unclear and warrants further research [154].

Kaempferol, another natural product of the flavonol sub-family, is known to possess anti-inflammatory, anti-cancerous, and anti-diabetic properties. Kaempferol has been shown to induce CD4^+^FoxP3^+^ T_regs_ generation, reduce IL-17A^+^CD4^+^ or RORγt^+^CD4^+^ cell frequency, suppress mRNA expression of proinflammatory cytokines, and increase FoxP3 gene expression in IMQ-induced psoriatic mice. Mechanistically, kaempferol inhibits the mTOR and NF-κB signaling [150].

Psoralen is a naturally occurring compound derived from the seeds of *Psoralea corylifolia* and other plants. The combination of Psoralen and ultraviolet A (PUVA) has proven to be a very effective photochemotherapy with potent antipsoriatic efficacy, despite the fact that the molecular mechanism responsible for the effects is incompletely understood [129]. Since dysregulated immune mediators released in inflammatory disorders are known to activate the mTOR pathway, Shirsath et al. examined the role of the mTOR pathway in the K5.hTGFβ1 transgenic (TG) mice model in the presence or absence of PUVA. They demonstrated that the PI3K/mTOR cascade, downstream effectors (e.g., pS6), and the upstream regulator EGFR were highly overexpressed in the diseased skin of K5.hTGFβ1 TG mice compared to wild-type mice. Moreover, treatment with PUVA significantly reduced Akt/mTOR activity in the TG mice as evidenced by the suppression of pS6 (Ser235/236) as well as a significant induction of Foxp3+ regulatory T (Treg) cells in the lymph nodes and skin of K5.hTGFβ1 mice [129,155]. The mechanism of how PUVA treatment leads to the downregulation of pS6 without having any effect on mTOR itself remained to be further determined. Therefore, analysis of skin biopsies for localization of phosphorylated S6 expression could determine the pharmacodynamics and efficacy of anti-psoriatic therapy [129,155,156,157].

##### Systemic Sclerosis

Systemic sclerosis (SSc), also termed scleroderma, is an acquired connective tissue disease that causes fibrosis of the skin and other organ systems [158]. SSc is clinically categorized into limited cutaneous SSc (lcSSc) and diffuse cutaneous SSc (dcSSc); the latter is more rapidly progressing with visceral involvement [159]. The pathogenesis of SSc is incompletely understood [158]. However, the underlying triggering factors are predicted to involve inflammatory response, injury to the endothelial cells, extra-cellular matrix (ECM) hyperplasia, and abnormal activation of the immune system or damage to vasculature leading to fibroblast activation as well as abnormal ECM accumulation [158]. Additionally, an abnormally thickened dermis resulting from excessive and uncontrolled ECM deposition is another hallmark of SSc [160,161,162]. In contrast to fibroblasts derived from healthy skin, cultured fibroblasts derived from SSc patients have been reported to overexpress type I collagen [161,163]. Moreover, upregulation of mTOR expression has been reported in tight skin (TSK+) mouse fibroblasts [164]. In corollary, Zhu et al. reported the overexpression of phosphorylated mTOR in the fibroblasts derived from patients with SSc and BLM-induced mice [159].

Insulin-like growth factor binding protein-5 (IGFBP-5) has also been identified in cultured dermal fibroblasts derived from SSc patients, which was established as a trigger of fibrotic phenotype through the induction of ECM assembly, the transformation of fibroblasts to myofibroblasts, as well as the infiltration of inflammatory mononuclear cells to the sclerotic skin lesion [165]. Moreover, IGFBP-5 has been reported to target the downstream signaling molecule docking protein-5 (DOK-5), a member of a class of proteins that are also known as insulin receptor substrates (IRS). In cutaneous SSc, IRS proteins act as specific signal transduction molecules and play an important role in cellular differentiation through mTORC2 [165]. Still, in SSc, immune-activated fibroblasts express α-smooth muscle actin (α-SMA) and are transformed into myofibroblasts which is responsible for the excessive production of ECM.

Several other triggering factors including platelet-derived growth factor (PDGF), transforming growth factor β (TGF-β), reactive oxygen species (ROS), and tissue hypoxia contribute to the persistent activation of fibroblasts [166,167]. TGF-β is the major critical mediator of fibroblast activation that has been associated with Akt activation in different SSc models [168]. Inflammatory mediators like TNF-α found at the lesioned site of systemic sclerosis patients have also been reported to play a vital role in SSc [168]. These mediators exacerbate SSc by phosphorylating mTOR and its downstream effector molecules S6K and S6, indicating that activation of the mTOR pathway plays a crucial role in the initiation of inflammatory cascade manifested in cutaneous SSc [159]. Moreover, infiltratory B cells in the dermis of SSc patients produce pro-inflammatory and pro-fibrotic cytokines (e.g., IL-6), resulting in increased expression of TGF-β, which all together causes fibroblast activation and ECM deposition (e.g., collagen) in the sclerotic lesion [169].

##### Targeting the PI3K/Akt/mTOR and Allied Networks for Treating Systemic Sclerosis

A major hallmark of systemic sclerosis is the remodeling of the connective tissue in the skin. A primary cause of this remodeling may be the lack of blood flow to the affected area, which causes the activation of fibroblasts and the upregulation of cytokine-induced connective tissue growth factor. Since the involvement of the Akt/mTOR pathway and its associated effector proteins have been reported to be linked to the pathogenesis of SSc, targeting inflammation and the Akt/mTOR and associated pathways is a promising approach in treating patients with this skin condition [159]. In a proof-of-concept study, the administration of rapamycin, an mTORC1 inhibitor, significantly prevented skin fibrosis in bleomycin-induced mouse models [163]. It is known that activation of Akt by mTORC2 limits the effectiveness of mTORC1 inhibition by rapamycin. Therefore, simultaneous targeting of both mTORC1 and mTORC2 to effectively block the PI3K/Akt/mTOR pathway seems imperative, and it is vital, as such a strategy has led to the discovery and development of new second-generation inhibitor molecules against SSc.

Because a characteristic feature of SSc is fibroblast activation that results in fibrosis and inflammation, Liang et al., studied the anti-fibrosis activity of BEZ235 (dual inhibitor of PI3K/mTOR) and rapamycin (mTORC1 inhibitor) in activated fibroblasts and in two murine models of SSc. By analyzing the levels of phosphorylated Akt, GSK-3β, mTOR, and S6, they demonstrated an upregulation of the PI3K/Akt/mTOR signaling components in primary dermal fibroblasts. Compared to rapamycin, BEZ235 more significantly inhibited fibroblast activity [170]. In the in vivo murine SSc models, BEZ235 treatment resulted in the vertical inhibition of dermal fibrosis associated with the inhibition of dermal thickness and collagen deposition [170]. In this light, a dual inhibitor of PI3K/mTOR, BEZ235 is now undergoing clinical development for SSc [170]. Since TGF-β receptor (TGF-βR) activation is one of the potent triggering factors in SSc pathogenesis, agents that can inhibit the TGF-βR along with dual mTORC1/2 inhibition could be effective in intervention against SSc. A recent study by Mitra et al. reported that OSI-027 (a dual mTORC1/2 kinase inhibitor) effectively inhibited the pro-fibrotic effect of TGF-β and PDGF, suggesting that it is a promising anti-sclerotic agent [171]. Cong et al., utilized AZD8055, a dual inhibitor of mTORC1/2, and showed a potent inhibitory effect on TGF-βR (IC_50_ = 86 nM), hence being a promising antisclerotic agent [172].

Sirtuin 1 (Sirt 1), an NAD^+^-dependent class III histone deacetylase, is a known regulator of cell growth and fibrosis and is important in the regulation of TNF-α induced inflammation and angiogenesis as exemplified by the fact that mice lacking endothelial Sirt1 display abnormal angiogenesis [159,173]. The ability of Sirt1 to curtail inflammation is an indication of its therapeutic potential against SSc. Zhu et al. demonstrated that activation of Sirt1 mitigated the expression of inflammatory and fibrogenic factors and also significantly suppressed the mTOR signaling in BLM-treated fibroblasts, suggesting that activation of Sirt1 may inhibit inflammation and ameliorates fibrosis in SSc patients via blocking the mTOR pathway [159].

Alternative natural herbal medicine approaches have long been utilized in different parts of the world in the management of human diseases and have recently been employed in the management of SSc as well. For instance, geniposide, a novel compound isolated from the traditional Chinese herbal medicine Zhizi, was shown to inhibit the mTOR pathway in bleomycin-induced SSc models [174]. Excessive levels of collagen expression and fibroblast activation occur in the advanced stage of SSc, which creates a hypoxic condition leading to activation of hypoxia-inducible factor 1 alpha (HIF-1α) further aggravating the situation. As a potent inhibitor of HIF-1α, 2-methoxyestradiol was found to be promising for treating SSc. 2-methoxyestradiol inhibited the fibroblast proliferation and induced apoptotic death in cultured cells. The mechanism of the antifibrogenic effect of 2-methoxyestradiol in scleroderma is exhibited by the inhibition of the PI3K/Akt/mTOR/HIF-1α signaling pathway since this axis becomes activated under hypoxic conditions. 2-Methoxyestradiol was found to exert this effect via decreasing the phosphorylation of Akt and mTOR and subsequently downregulating HIF-1α expression [175].

##### Lichen Planus (LP) and Oral Lichen Planus (OLP)

Lichen planus (LP), is a common chronic inflammatory dermatological disease affecting mucocutaneous tissues such as the skin, and oral and genital mucosa, and is known to affect adults older than 30 years, with incompletely understood etiology and no known curative treatment [176,177]. Histologically, cutaneous lesions of LP are self-limiting, and are characterized by the destruction of epithelial basal cells associated with the formation of apoptotic bodies and sub-epidermal band-like infiltration of lymphocytes [178,179].

Oral lichen planus (OLP), is the chronic inflammatory disease variant of LP that solely affects the mucus membrane of the oral cavity, presenting an overall prevalence of 0.5–2% within the adult population, with a significant 0.4–5% rate of malignant transformation, and it essentially affects women of middle age [180,181]. Despite self-limiting, once OLP is established, the lesions are non-remissive and are associated with exacerbating cyclic and quiescence periods and sometimes with increased malignant potency [182,183]. Furthermore, degeneration of keratinocytes with intraepithelial and subepithelial infiltration of mononuclear cells are major histological features of OLP, since keratinocytes play a vital role in the normal integrity of the skin. The etiology and molecular mechanisms of OLP pathogenesis are complex and not fully elucidated. OLP lesions are demonstrated in six clinical features- three of which are categorized as white keratotic lesions, which include plaque-like, popular, or reticular lesions, while the other three are white keratotic lesions accompanied by red sores named bullous, atrophic, or erosive OLP [184,185]. Because of the possible transformation of OLP into a malignant complication in the long term, which may lead to squamous cell carcinoma. The World Health Organization (WHO) categorized OLP as a potential premalignant lesion [186].

Though the obvious etiology of OLP remains to be determined, a multifactorial disease nature is suggestive including predisposing factors such as genetic, environmental cues like stress, trauma, and microbial factors, including bacterial, fungal, and viral infections which nonetheless according to established guidelines, is yet to be established as causative factors in OLP pathogenesis [179,180,181,182]. Some studies have reported abnormal immune activation, which stimulates dendritic cells to secret cytokines such as IL-12, IL-18, TNF-α, etc. that recruit and differentiate T cells to Th1 and Th17 cells in the site of the lesion. These cells in turn secrete other inflammatory cytokines (e.g., IL-6, IL-8) leading to OLP generation [187]. Nonetheless, current literature considers that OLP is a T-cell mediated autoimmune disease where cytotoxic CD8+ T cells are responsible for triggering the observed apoptosis of the basal keratinocytes of oral mucosa epithelium [183,188]. The characteristics of pathogenesis now mainly include antigen-specific and nonspecific inflammatory mechanisms. Here, the antigen-specific mechanisms comprised of activation of T-cells subsequent to antigen presentation and basal keratinocytes apoptosis triggered by CD8+ cytotoxic T cells, while non-specific mechanisms involve over-expression of matrix metalloproteinases (MMPs) and degranulation of mast cell in OLP lesions [181,182,183,188].

Recent studies have demonstrated the activation of Akt/mTOR signaling pathway components in the local OLP tissues of patients with erosive OLP, compared with controls [181,189]. It was observed that the inflammatory mediators released by the infiltrated mononuclear cells in OLP activate the PI3K/Akt/mTOR signaling pathway, which causes the elevated level of MMP-2. MMP-2 is responsible for the proliferation and apoptosis of keratinocytes exhibiting the clinical features of OLP [182]. Yet, in another study, higher levels of p-Akt, mTOR, and ribosomal protein S6 (pS6) were observed in patients with OLP. As a downstream target of mTOR, pS6 upon phosphorylation induces cellular growth and proliferation and participates in protein synthesis. In the report, activation of the Akt/mTOR/pS6 signaling pathway was shown to indicate the premalignant potential of OLP to the development of oral squamous cell carcinoma (OSCC) [189]. Because MMP-2 is responsible for the keratinocytes displaying clinical features of OLP, MicroRNA-125b (miR-125b) has been shown to reduce the level of activated Akt and mTOR, thus reducing MMP-2, which ultimately decreases LPS-induced keratinocyte proliferation and induces apoptosis [182]. Activation of Akt/mTOR signaling was found in the OLP patients, which may be contributed to the premalignant potential of individual cases [182]. Zhang et al., data showed increased expression of phosphorylated Akt and mTOR in OLP local tissues, which actually indicated the activated Akt/mTOR-autophagy in OLP lesions [181]. Moreover, Prodromidis et al., performed immunohistochemistry to analyze the activation status of the Akt/mTOR signaling pathway in OLP. They found that almost all normal mucosa had no Akt/mTOR activation at all in the cytoplasm, but the Akt/mTOR signaling pathway was significantly activated in OLP patients’ lesions [189]. This happened due to the different stages of the disease course and different inclusion criteria. More interestingly, it has been suggested that Akt/mTOR activation occurs in the context of OLP patients. A study by Wang et al. demonstrated that miR-125b lowered the enhanced phosphorylation of Akt protein and its downstream target, mTOR protein, in LPS-incubated HaCaT cells, highlighting the potential contribution of PI3K/Akt/mTOR signaling pathway aberrations in potential OLP malignancy [182].

##### Targeting the PI3K/Akt/mTOR and Allied Networks for Treating LP and OLP

Available treatment options for OLP range from topical corticosteroids to laser ablation of the lesion, but they are mostly symptomatic, and none are curative as the exact etiology and pathogenesis of OLP are unclear, which presents a major hurdle toward the development of novel mechanism-based therapies [177,179,181,182,188]. Since activation of mTOR signaling occurs in OLP, targeting mTOR has been regarded as a promising approach in OLP treatment. In a case study, two patients with refractory LP were reported to have successful remission of the disease with oral sirolimus (rapamycin) treatment [190].

MicroRNAs (also miRNAs), are highly conserved ~19–22 nucleotides long single-stranded noncoding, and are important in the development, morphogenesis, and maintenance of life through their influence on cell growth, differentiation, apoptosis, autophagy, immunity, and inflammation; in human skin, miRNAs are involved in diverse skin disorders including AD, psoriasis, and cancers [191]. Cosmeceutical use of molecules/natural active ingredients to regulate miRNA expression for significant advances in skin health/care product development was not recognized until recently [191]. Regulation of miRNA expression is a promising approach for novel skin disease target discovery and therapeutic interventions that could enhance skin health. The miRNAs mediate post-transcriptional silencing through partial complementary binding to the 3′un-translated region (UTR) of the target mRNA. Decreased expression of some miRNAs has been observed in immune-mediated inflammatory skin lesions including OLP [192,193,194] psoriasis [194,195,196], and atopic eczema [197,198,199,200,201] compared to healthy non-lesional skin. In OLP, Wang et al. employed HaCaT keratinocytes stimulated with lipopolysaccharide (LPS) to mimic the OLP immune environment and examined, under OLP-like settings, the role of miR-125b in the growth and apoptosis of keratinocytes. They demonstrated that miR-125b inhibited proliferation and promoted keratinocyte apoptosis by targeting MMP-2 expression through PI3K/Akt/mTOR pathway [182], and altered expression of miR-21, miR-125b, and miR-203 specifies the role of these miRNAs in OLP [201]. They demonstrated that the upregulation of MMP-2 expression was concomitant to a down-regulation of miR-125b in both OLP mucosa tissues in vivo and LPS-stimulated HaCaT keratinocytes. Moreover, Overexpression of miR-125b was observed to inhibit LPS-induced MMP-2 expression and cell proliferation and enhance apoptosis in HaCaTs associated with the activation of Akt/mTOR [182].

##### Acne Vulgaris

Acne vulgaris (AV) refers to a common chronic inflammatory skin disorder that affects most people during some point in their lives, especially during the stage of pubertal development, and includes about 85% of adolescents in Westernized populations [202,203,204,205] AV is a multifactorial condition associated with the obstruction and inflammation of the pilosebaceous follicular units that can be related into four different processes including sebum overproduction, abnormal follicular epithelium shedding, Cutibacterium acne or Propionibacterium acnes colonization, and inflammation and is characterized by comedones, oily skin, and cysts [202,203,205,206,207]. Though the processes may occur in any order at any time, many researchers are interested in the inflammation process since it has been noted to occur in all stages of the disease [208,209,210]. The exact mechanism underlying AV development is not fully understood but several major factors such as hyperkeratinization of the pilosebaceous follicle, colonization of the follicular duct by *Propionibacterium acnes* (*P. acnes*), increased sebum production, and release of inflammatory mediators in the skin have been demonstrated to be involved in AV pathogenesis [208,209,210]. Thus, increased synthesis of protein and sebaceous lipogenesis, sebaceous gland hyperplasia, cell proliferation and differentiation of acroinfundibular keratinocytes, insulin resistance, and body mass index are associated. Other factors that contribute to the development and persistence of AV include hormones; psychological stress; genetics; and certain food and drink intake, such as high glycemic index, saturated fat with omega-3, and omega-6 fatty acids diets (which increase the level of insulin and insulin-like growth factor 1-IGF-1), tobacco smoke, and damaged or unhealthy skin [203,207,208,211,212,213]. AV shows a higher prevalence in urban populations compared to rural areas illustrative of a more western lifestyle [203,208]. A detailed understanding of the molecular target and factors that regulate sebum production and inflammatory processes is important for the discovery and development of novel potent molecular target-based treatments for acne. In this light, while both insulin and IGF-1 have been shown to stimulate sebaceous gland lipogenesis, IGF-1 has been reported to induce the expression of sterol response/regulatory element binding protein-1 (SREBP-1), a transcription factor that regulates a number of lipid biosynthesis genes. The upsurge of SREBP-1 expression stimulates lipogenesis in sebocytes as evidenced by the fact that treatment of SEB-1sebocytes with IGF-1 resulted in the induction of SREBP-1 expression and lipogenesis via the modulation of PI3/K/Akt/mTOR/FoxO1 signaling pathway [95,206]. This suggests that in sebocytes, IGF-1 transmits its lipogenic signal through the activation of Akt, meaning that specific pathway-targeted modulation in the sebaceous gland could represent a suitable strategy for curbing sebum production and improving acne [96,205]. Since the PI3K/Akt pathway is an upstream modulator of mTORC1, and Akt is a downstream target of mTORC2, it is evocative that the mTORC1 pathway likely plays a crucial role in the pathogenesis of AV [214]. Additionally, insulin, IGF-1, and other growth factors are able to activate the nutrient-sensitive mTORC1, which is known to be associated with different types of skin lesions. Moreover, mTORC1 has also been reported to be associated with adipogenesis and lipogenesis by activating the SREBP-1, which is responsible for stimulating sebaceous gland fat production/lipogenesis [96,204].

Increased expression of mTOR and its downstream target S6K1 has been reported in skin lesions of AV patients compared to individuals with normal skin [215]. Recently, it has been suggested that deficiency of the forkhead box transcription factor O1 (FoxO1) is linked to the pathogenesis of AV; an increase in phosphorylated FoxO1 (p-FoxO1) in the cytoplasm of cells derived from patients with acne is consistent with activation of mTORC1. FoxO1 is phosphorylated and inhibited by Akt, and it acts as an important rheostat regulating the activity of Akt and mTORC1 while acting as a potent inhibitor of mTORC1 [204,214,216,217]. This strongly indicates that mTORC2 activation is critical for the development of acne, considering that the activated Akt by mTORC2 phosphorylates FoxO1, which promotes the nuclear-cytoplasmic translocation and the degradation of FoxO1, thus resulting in FoxO1 deficiency [96].

##### Targeting the PI3K/Akt/mTOR and Allied Cascades for Treating AV

Due to the multifactorial nature of AV, treatments are more challenging, and there is no available standardized mechanism and target-based therapy. Current management and care for AV include the usage of topical retinoids, benzoyl peroxide, antibiotics, azelaic acid, oral zinc, isotretinoin, dapsone, taurine bromamine, resveratrol, chemical peels, optical treatments, and alternative and complementary medicine [218,219]. Besides, novel molecules with little resistance potential include octadecenedioic acid, phytosphingosine, lauric acid, retapamulin, T-3912, and NB-003. The use of oral retinoids and non-antibiotics such as zinc can inhibit resistance as well as relief on antibiotic dependency [220].

Given the role of the Akt/mTOR network in AV, there is a great need to identify and develop novel drugs targeting the mTORC1- and FoxO1-dependent signaling pathways to mitigate mTORC1/SREBP1-mediated sebum production as treatment options [221]. Several natural dietary product supplements have been described as anti-acne agents, which target the mTOR signaling pathway. One of the most widely used compounds is isotretinoin (13-cis-retinoic acid), which is considered a gold standard and first-line treatment for severe AV [222]. It is the most powerful sebum suppressor, which acts by increasing the nuclear expression levels of FoxO1, thus inhibiting mTORC1 leading to apoptotic death of sebocytes [223]. As mentioned before on the relation of diet on AV, several natural dietary product supplements are also being used for a sustained lifestyle adaptive treatment regime that targets the mTOR signaling pathway. Natural dietary polyphenolic ingredients like EGCG and resveratrol have been reported as potent mTORC1 inhibitors, which inhibit sebaceous fat production while reducing clinical appearances of AV reviewed in [206]. In a pilot study, topical application of a 0.01% *W*/*V* resveratrol-containing gel, significantly decreased the clinical signs of acne in patients with facial AV [224]. Moreover, the growth inhibitory effect of resveratrol was enhanced by treatment with LY294002 (a potent PI3K inhibitor), suggesting an inhibitory mechanism associated with Akt inhibition, and thus its therapeutic use for treating AV [225].

Vitamin D deficiency has been observed in a large number of individuals with AV, and both keratinocytes and sebocytes express the vitamin D receptors (VDR), which subdue cell growth through mTORC1 inhibition, therefore vitamin D supplementation is suggested to be an attractive therapeutic strategy for AV [226]. Recently, metformin, a known antidiabetic and anticancer agent has garnered interest for use in the management of dermatological disorders linked with insulin resistance such as AV, since it was identified as a dual inhibitor of mTORC1/2, thus having the potential to ameliorate the conditions in AV reviewed in [206,224]. Metformin alone or in combination with various diets has been shown to pre-clinically and clinically suppress SREBP levels in pre-clinical mice models and clinically in patients with acne [206,227]. The multicomponent Chinese herbal medicinal preparation Compound Muniziqi granule (MNZQ), which contains 13 medicinal plants has been reported to be useful in the treatment of endocrine disorder-induced acne as well as chloasma and others [228]. An alternative therapy for AV involves photodynamic therapy with aminolevulinic acid (ALA-PDT), which has been found to inhibit mTORC1 with subsequent inhibition of downstream signaling molecules (e.g., S6K, etc.) in human SZ95 sebocyte cells. Therefore, ALA-PDT blocks the Akt/ERK/mTOR-S6K signaling cascade, and reduce the number of sebaceous gland through ROS-mediated cell membrane damage and apoptosis induction to sebocytes [229]. Others such as platycodin D, lupeol, etc., have been identified for their anti-acne effect through their inhibition of the PI3K/AkT/mTOR and associated pathway [211,212,228].

##### Hidradenitis Suppurativa

Hidradenitis suppurativa (HS), also termed acne inversa is a debilitating and chronic inflammatory skin disease (CISD) that can be viewed as an auto-immune keratinization disorder primarily affecting apocrine-gland-rich (AGR) areas of the body [230,231,232]. Clinically, HS is characterized by a relapsing palpable painful lump, which can aggravate into a leaky pus sore (abscess), most commonly found in the groins, armpit, buttocks, breast, or apocrine-producing regions [233]. The thin contrast between HS and other common skin disorders such as acne vulgaris, boils, and herpes, makes misdiagnosis a common occurrence. However, both HS and AV result from the occlusion of the follicular-pilosebaceous unit; however, unlike AV, where sebum, dead cells, and dirt have been well-associated with this, keratin and risk factors such as obesity and smoking have been identified to hike HS incidence [234]. Though the molecular basis of HS pathogenesis is incompletely characterized, it is a multifactorial process that culminates in the obstruction of the follicular-pilosebaceous unit ignites a chronic inflammatory cascade and can be worsened by opportunistic pathogens (*Staphylococcus lugdunensis, Streptococci* spp., and *C. albicans*) though the condition itself is non-infectious. HS has a vicious circle inflammatory cascade that ultimately leads to pain, purulence, tissue destruction, and scarring. Deregulation of the innate and adaptive immune systems, inflammatory mediator orchestration, other factors such as genetics, hormonal status, obesity and smoking, lifestyle, as well as diseases like psoriasis have been implicated as risk factors associated with the pathogenesis of HS. Moreover, HS is more prevalent in females than males and rarely occurs before puberty, especially among African Americans, Hispanics, or people of interracial descent [230,235,236]. This immune-mediated disorder has been shown to involve the dysregulation of the gamma-secretase/notch pathway, which is needed to regulate the activation of the innate immune cells by MAPK specifically p38, JNK, and ERK1/2 by increasing the biosynthesis of pro-inflammatory cytokines such as TNF-α and IL-1β [237,238]. We and others have shown that the mTOR signaling pathway, especially the activation of the mTORC1 might be linked to the pathophysiology of HS [239,240,241], and could predict treatment response outcomes with Isotretinoin [223].

##### Treatments Strategies for HS

There is no absolute cure for the HS condition, and due to the wide range of its clinical manifestations and the complex pathogenesis, the therapeutic approach has been challenging. However, pharmacological interventions including topical, systemic, biological agents, surgery, and light therapies used for palliative management of affected patients work through the mitigation of these dysregulated immune mediators, cytokines, and signaling pathways with variable clinical outcomes [242]. Examples include TNF inhibitors; adalimumab and infliximab, hormonal therapy, retinoid, triamcinolone, and many more. Laser therapy, surgical removal, incision and drainage, lifestyle modification such as smoking cessation, controlling obesity, and dieting are all useful for HS management [235]. We have recently shown the anti-inflammatory effect of Rifampicin in ex vivo cultures of HS [243].

##### Alopecia Areata

Alopecia Areata (AA) is an autoimmune disorder where immune cells attack hair follicles. It has a non-sex-linked lifetime risk of about 1.7% and a prevalence of 0.1–0.2% among the general population. It is commonly seen as a small rounded patch or coin-sized hair loss on the scalp that is accompanied by anxiety, depression, and low self-esteem among some affected individuals [244,245]. When compared to healthy individuals, AA patients were found to have less immune privilege in their proximal anagen hair follicles. This loss of ability at the hair follicles precipitates NK cells (mostly elevated among such individuals) to attack the intra-follicular expressed auto-antigen, degenerate its vacuole, shortens, weakens and eventually leads to hair fallout evident with prolonged telogen but hasty anagen phase [246,247,248].

The role of the PI3K/Akt/mTOR and associated pathways in AA is seen by the connection they share with biological markers characteristic to this condition such as NK cells, CD4+, CD8+, MHC classes I and II cells, and migration inhibitory factor (MIF). Class 1A PI3K intracellular signal transmitting enzymes p110α and p110δ have been identified to play a role in lymphocyte chemotaxis and migration as shown using a knockout mice model which resulted in a significant reduction in NK number in peripheral organs, phenotypic anomalies, and impaired cytokine secretion in other cells [249]. Furthermore, AA patients tend to have an increase in MHC classes I and II cells and a decrease in MIF, which normally suppresses the signaling for the transport of CD56+/NKG2D+NK cells toward the hair follicle. However, when MIF is decreased, these cells can attack the hair follicle and weaken it [1,4]. CD4+, CD8+, and other immune mediators such as macrophages, Langerhans’s cells, and cytokines have all been implicated in the development and progression of the condition [245,247,248,250].

While AA is primarily an autoimmune disorder, stress, infections, and environmental elements may also be a predisposing factor. In fact, patients diagnosed with AA have been found to have an increase in oxidative biomarkers including malondialdehyde and superoxide dismutase (SOD) and antibodies that are known to target oxidative species within the body [247,248].

The condition can be self-limiting but, in some instances, agents that are postulated to affect the PI3K/Akt/mTOR, its downstream targets, and other pathways are used. However, the defiance associated with the management of this condition lies in its relapse propensity and fickle nature in affecting previous unexposed sites. Potential agents for AA treatment include small molecules such as corticosteroids; immunomodulators; and natural products such as sibilin, caffeine, capsaicin, curcumin, garlic gel, onion juice, melatonin, procyanidin, pumpkin seed oil, rosemary oil, Saw palmetto, vitamin B7 (biotin), Vitamin D, Vitamin E derivatives, and zinc [251,252,253,254].

##### Buruli Ulcer

Buruli ulcer (BU) is a chronic necrotizing noncontagious infectious skin disease caused by an atypical mycobacterial pathogen, *Mycobacterium ulcerans*, an environmental bacterium, and BU constitutes the third most common mycobacterial disease worldwide, with incompletely understood disease mechanism [255,256]. The pathogenicity of *Mycobacterium ulcerans* is centrally linked to its production of a macrolide exotoxin termed Mycolactone, a unique member of the mycobacteria species [257,258,259]. The disease mechanism in BU is incompletely understood, and the incidence has been reported in over 33 countries worldwide including tropical African countries, and the mode of transmission of *M. ulcerans*, as well as its transmission vector still remains elusive [255,256]. Perforated skin site due to insect bite or trauma is anticipated to be a potential route of disease transmission [260], and the disease’s clinical appearance is manifested as a severe ulcerative sore on the skin. The initial disease stages are signified as plaques, papules, edema, and nodules, where the nodules then undergo necrotic cell death leading to ulceration. While the cascade of events is painless without presentation of any systemic symptoms [261], BU may cause extensive skin and soft tissue damage that may even affect the bones if left unattended, and the scars formed from long-term BU infection may cause debilitating deformities of the vital organs [262].

The causative factor of BU mycolactone, is a cytotoxin, and immunosuppressive agent which acts on the same intracellular target (mTOR) as rapamycin to interfere with the assembly and inhibit the mTOR pathway. The cytotoxic effect of mycolactone is due to the inhibition of the rictor-containing mTORC2 to prevent Akt phosphorylation, which consequently leads to the dephosphorylation and activation of the Akt-targeted transcription factor FoxO3 and upregulation of FoxO3 target gene BCL2L11 (Bim) and thus apoptotic cell death via the mTORC2-Akt-FoxO3 signaling axis [263,264,265,266,267,268,269]. The WHO recommended that treatment for BU lesiona less than 10 cm in diameter is daily oral rifampicin and intramuscular streptomycin or a combination treatment with rifampicin plus clarithromycin, ciprofloxacin or moxifloxacin. For larger lesions, surgery is recommended [263]. There is currently no treatment option for BU that targets the mTOR pathway, a major target for the exotoxin of the bacteria mycolactone, indicating this is a potential therapeutic target for novel drug development.

##### Wound Healing, Hypertrophic Scars, and Keloids

As discussed in the preceding sections, the skin has numerous unfathomable roles resistive/reliant on a compact morphological barrier layout that is required for the execution of homeostasis (see Figure 1 and Figure 2). Wounds or injuries that may result from external physical or chemical abrasion/shock to the body tissue tend to compromise these structural barrier layouts of the integumentary system. Wounds could also be referred to as skin ulcers if it has internal etiology, which in many instances makes them chronic in nature and requires extensive medical care [265]. Wound healing is a natural physiological response of the body, which has an inverse relation with aging, with older patients experiencing impaired wound healing. Despite it being an intrinsic mechanism of the body, the large size of wounds, malnutrition, medical conditions like infections (e.g., buruli ulcer see the section above), metabolic disorders (e.g., diabetes), genetic disorders, cancers, and other conditions can distort its functionality [5]. It is reported that about USD 25 billion in healthcare expenditure is expended for the treatment of chronic wound, with an estimated USD 15 billion commercial market value for wound care products, an indication of how expensive wound care significantly contribute to national budget usage in the U.S. alone [265]. For instance, in second to third-degree burn sufferers, patients often received chemotherapy and/or radiotherapy, while the elderly, and patients with diabetes often experience slow or recalcitrant healing of ulcers, and these pose major clinical huddles, and thus a significant public health concern [266].

There are four main stages or phases normally involved in the process of wound healing, which include the hemostasis phase, inflammatory phase, proliferation phase, and tissue remodeling phase. These phases work in a sequence connected network where each phase subsequently leads to the other, all commencing with hemostasis and inflammation phases following skin injury then climaxing with the remodeling phase, which exclusively maintains immunity and the fluid balance of internal tissues within 0–72 h duration. The processes are characterized by vasoconstriction and the initiation of tissue aggregation through the release of clotting factors, growth factors, and chemotactic factors mainly through the activity of thrombocytes/platelets. Subsequently, vasodilation occurs, which facilitates the migration of leukocytes predominantly the neutrophils, macrophages, and monocytes to the injury site by chemokines, to get rid of wound debris and bacteria via phagocytosis [267,268,269].

The anatomic viewpoint of skin regeneration reveals that inner-layer cells migrate to replace outer-layer cells, and this proliferative phase increases the threshold of the phenomenon and begins about 3–10 days after injury. In the proliferative or remodeling phase, new tissues are formed to cover the wounds, and because these tissues must be sustained with the required nutrients and oxygen supply hence angiogenesis features occur at this stage. The granulation tissues are formed from collagen and extra-cellar matrix. Several studies on re-epithelization and other processes related to wound healing have mainly served as the basis for therapeutic exploits of some complications associated with wound healing [270,271]. Epithelial-mesenchymal transition (EMT), a component of re-epithelization, where the epithelial cells lose their coherent bonds and change to mesenchymal stem cells is enhanced by many triggering factors among which TGF-β plays a vital role [272]. Considering the downstream mechanism of the TGF-β as related to EMT, Smad, RhoA, MAPK, and, most importantly for our review, the PI3K/AKT/mTOR have been successfully identified as promising molecular targets for treating wounds [272].

The remodeling phase notable for the transformation of collagen III to collagen I and the abruptions of angiogenesis from the proliferation phase, culminate the entire wound healing process and starts about 14–21 days after injury. Tissue granulations also terminate through apoptosis of excess cells, making way for the strengthening of the formed tissues. Type I collagen formation tends to restore the tensile strength of skin but, in many instances, hair follicles and sweat glands function are lost in severe wounds [273].

Hypertrophic scars and keloid scars are complications of excessive wound healing. These are benign dermal fibro-proliferative lesions with a high repetition rate in postsurgical conditions mainly occurring in genetically susceptible individuals, thus exposing the essence of proper control in the healing process through modulatory perspectives [274,275,276]. Several studies have revealed that genetic factors are associated with excessive wound-healing maladies, especially among African Americans and Asians. In addition, poor transition between one of the major phases to the other leads to complications of excessive wound healing. Moreover, primary keloid-derived fibroblasts have been found to display some “cancer-like cell” characteristics, though keloids are not truly malignant neoplasms considering that keloids do not metastasize [276,277,278,279]. Hypertrophic scars predominantly contain a parallel arrangement of collagen III whilst keloids have both collagens I and III but in a disoriented fashion with skin surface [269,275]. Importantly, components of the major mTOR signaling pathway have been shown to control the expression of collagen I in human dermal fibroblasts through a PI3K-independent mechanism [276], but the PI3K/Akt/mTOR pathway has been found to be responsible for excessive production of collagen as well as ECM in keloids [280,281,282,283]. In addition, increased angiogenesis and immune cell infiltration are observed in keloids, suggesting the involvement of mTOR pathway, which is a plausible therapeutic target for the management of keloids [276].

##### Targeting the PI3K/Akt/mTOR and Allied Cascades for Treating Wound Healing, Hypertrophic Scars, and Keloids

Many approaches are utilized to boost wound healing; both in allopathic or orthodox, and alternative medicine. In the orthodox approach, pharmacological agents such as iodine, penicillins, sulphadiazine, neosporin, polysporin, medicated bandages, etc., are utilized in wound care and management. Some instances, especially with major wounds may require non-pharmacological means where skin grafting is considered. Because the PI3K/Akt/mTOR pathway enhances the proliferation of cells involved in the reepithelization of wounds, it is probably a critically important pathway to be targeted for enhanced wound healing. Overexpression of activated Akt (Ser473) and S6K1 has been reported in the spinous through the granular layer in the transitional epithelium of mice incision skin wounds enduring healing, and downregulation of PTEN, a negative regulator of PI3K/Akt resulted in activation of Akt and enhanced wound healing [95]. Furthermore, ablation of TSC1 in mice resulted in augmented wound healing, in corollary another in vivo/clinical human study demonstrated increased wound healing times following treatment with everolimus, an mTOR inhibitor [95].

Honey, curcumin, ozone oil, etc., are employed in alternative medicine approaches, with some having modulatory effects against inflammatory and hyperproliferative properties by modulation of the PI3K/Akt/mTOR pathway. Curcumin used as a topical agent inhibits the NF-κB, PI3K/Akt, and IKK pathways [284]. In line with these assertions, a recent study by Xiao et al. demonstrated both in vitro and in vivo that ozone oil facilitates wound healing through the induction of fibroblast migration and EMT in a mechanism associated with modulation of the PI3K/Akt/mTOR signaling pathway, suggesting a novel therapeutic target for wound healing [272]. Moreover, treatment of wounds associated with Keloid scar formation by targeting PI3K/Akt/mTOR has been reported [276,277,278,279,280,281,282,283,284,285,286,287,288,289,290,291,292,293,294,295,296,297,298,299,300]. Rapamycin treatment has been shown to inhibit ECM deposition and a comparative study of rapamycin, P529, and wortmannin by Syed et al., demonstrated that P529 and wortmannin significantly suppressed keloid-associated fibrotic markers collagen I, fibronectin, and α-SMA in in vitro and ex vivo models. These suggested the importance of targeting the PI3K/Akt/mTOR axis for treating keloids [276]. In another study by Syed et al., they demonstrated that dual inhibition of mTORC1/2 by the agents KU-0063794 and KU-0068650 are promising therapeutic agents for the management of keloids [285].

##### Mycosis Fungoides

Mycosis fungoides (MF) or Alibert-Bazin syndrome is a rare cutaneous disease, which is a non-Hodgkin lymphoma but with a profound chronic skin inflammatory clinical presentation [286]. This condition has been tagged with the aged population and is estimated to affect about 3.6 per million people annually with males at a double-fold risk compared to females [287,288]. MF tends to progress slowly and many affected individuals suffer the health repercussions, which is akin to symptoms of conditions such as dermatitis, psoriasis, leprosy, and discoid erythematosus with profound patches which are scaly, itchy, flat, and reddish in nature. The shared similarities in signs and biomarkers have led to many patients being misdiagnosed especially during the early stages of the disease chronicity [289].

The exact cause of MF is not well understood and still being exploited. However; DNA alteration by either addition or deletion of chromosomes 7 and 17 or 9 and 10 respectively, is anticipated to have an influence [290]. Defect in the human lymphocyte antigen class II gene, cardinal to curb autoimmune response, has also been implicated in MF development (https://ghr.nlm.nih.gov/condition/mycosis-fungoides (accessed on 12 January 2023)). MF characteristically proceeds as patches, plaques, and finally as a tumor, which poses detrimental effects on other internal organs such as the liver, gastrointestinal tract, spleen, or brain in advanced condition. Patches can be either transient with relapses or persistent which are often seen on the lower abdomen down to the lower limbs and the breast. Despite the slow progression of the disease, patches can advance to plaques where the initially affected areas begin to exhibit lesions, which are itchy and reddish, purplish, or brownish in color (https://rarediseases.org/rare-diseases/mycosis-fungoides/ (accessed on 12 January 2023)). However, the plaques can also develop on a skin surface previously void of patches. Tumors are the terminal stage of MF and the disease condition derives its name “mycosis” from its morphological display: nodules as a mushroom—a fungi. The nodule results from cancerous T-cells, which are thicker and deeper than plaques that sometimes escalate to open sores [291].

##### Targeting the PI3K/Akt/mTOR and Allied Cascades for Treating MF

In spite of the turbid acumen on the pathophysiology of the MF, targeting the PI3K/mTOR pathway by inhibiting the phosphorylation of mTORC1 has shown a tremendous effect on cytokines, which affect the inflammatory response in this disease condition [292]. Everolimus was successfully used in vitro, in vivo, and in clinical trials to accentuate the influence of the PI3K/mTOR pathway, and also rationalize it as a target for therapies [293,294]. Topical preparations of corticosteroids, imidazoquinoline, mechlorethanamine hydrogen, bexarotene, carmustine, and tazarotene have been used for the early-stage treatment of MF [295,296]. Natural phytochemical agents such as curcumin and gartanin are reported to be linked with the treatment of MF via the inhibition of the mTOR downstream target S6K1 [96]. Phototherapy remains to be a classical therapeutic approach for the early phases of MF; psoralen–UV-A (PUVA) and narrowband UV-B (NBUVB) are mostly used [297].

##### Vitiligo

Vitiligo is a common autoimmune depigmentation skin disorder typically characterized by amelanotic lesions associated with non-scaly, chalky-white macules with distinct margins resulting from selective loss of cutaneous melanocytes with influences from genetic and environmental factors [298,299,300,301]. Vitiligo often affects an estimated 0.5–2% of the world’s population and causes a devastating psycho-social burden on the daily life of affected individuals and their families. Vitiligo is classified into two categories: segmental vitiligo (SV; exhibiting segmental patterns, rapid onset with leukotrichia (white hair)) and nonsegmental vitiligo (NSV; occurring in early ages with a halo nevus in families that have premature gray hair genes) [298].

The NSV is further categorized into generalized, acrofacial, mucosal, universalis, focal, and mixed, with generalized and acrofacial NSV being the two most common categories for detailed see review in [298].

The deregulated targets in vitiligo are known to be involved in immune regulation, melanogenesis, apoptosis regulation, and other disease states such as thyroid disease, type 1 diabetes, and rheumatoid arthritis. The most well-studied genes are the tyrosinase gene (TYR; limiting step for melanocyte biosynthesis and also found to be a major autoantigen in generalized vitiligo), NALP1 (found on chromosome 17p13 and encoding the NACHT leucine-rich protein 1), and XBP1P1 gene (x-box binding protein 1; commonly associated with vitiligo [301]. The protein tyrosinase is responsible for mitigating unfolded protein response (UPR) and, in some cases, drives a stress-induced response during in vivo studies [298]. Together, they interact to regulate part of the innate immune system.

Several cytokines, genetic factors, and environmental factors may also result in the pathogenesis of vitiligo and thus are important in the pathogenesis of vitiligo [301].

One of these protective mechanisms includes the α-melanocyte stimulating hormone (αMSH), which is responsible for several activating roles including surface receptor MC1R, inducing human melanocyte proliferation, and preventing human melanocytes from undergoing UV-induced apoptosis. Furthermore, αMSH is important in preventing oxidative stress from damaging melanocytes. For example, hydrogen peroxide normally damages and induces the loss of dendrites within human melanocytes, but αMSH prevents this damage from occurring [2,302].

The PI3K/Akt/mTOR pathway is vital for protection against oxidative stress often associated with the pathogenesis of vitiligo. For instance, Akt activates several growth factors that protect against UV radiation or oxidative stress-induced apoptosis, and it has been shown that αMSH activates Akt phosphorylation and the mTORC1 pathway. Moreover, exposure of cells to PI3K/Akt inhibitor (e.g., LY294002) or mTOR inhibitor (rapamycin), was reported to suppress αMSH activity by inhibiting α-MSH-induced dendrites. Thus, αMSH activates the PI3K/Akt/mTOR pathway, and its inhibition can protect against and treat vitiligo via protecting against ROS-induced oxidative stress [298].

Another study suggests that the PI3K/Akt pathway can be activated by the basic fibroblast growth factor (bFGF) to increase cell proliferation, migration, and differentiation of several types of cells including melanocytes. bFGF is released and derived from keratinocytes and helps melanocytes in migration. In fact, bFGF has been linked to increase melanocyte proliferation and migration. Patients with vitiligo showed a decrease in bFGF. Furthermore, the PI3K/Akt pathway is necessary for melanocyte migration in exposure to bFGF. When cells were exposed to LY294002, melanocyte migration was inhibited despite being exposed to an increase in bFGF. Without the pathway, bFGF cannot display its function, and melanocytes cannot migrate or proliferate correctly through exposure to bFGF [300,303].

The tyrosinase pathway, which starts with microphthalmia-associated transcription factor (MITF), is also connected to the PI3K/Akt pathway in vitiligo, as MITF is a regulator of melanocyte differentiation, pigmentation, proliferation, and survival, and is responsible for mediating tyrosinase as well as tyrosine-related protein 1 (TRP1) and TRP2. TRP1 is responsible for converting 5,6-dihydroxyindole-2-carboxylic acid (DHICA) into indole-5,6-quinone-2-carboxylic acid in mice, while TRP2 is more common in humans and is responsible for rearranging DOPA into DHICA [304]. Decreases in MITF in melanocytes are associated with a reduction in the melanocyte differentiation process and halting of melanogenesis. The PI3K/Akt pathway plays a significant role in the melanogenesis process via upregulating MITF and its regulated enzymes, as evidenced by the reduction in melanin synthesis upon PI3K pathway inhibition [301].

Akt is pivotal in the Akt/GSK-ß/ß-catenin pathway, as activated Akt phosphorylates GSK-3ß, which is required for stabilizing ß-catenin. ß-catenin plays an important role in the MITF pathway, as it upregulates MITF expression through its binding to the promotor region on the MITF gene. Upregulation of MITF promotes the expression of the TYR, TYRP-1, and TYRP-2 leading to an increase in melanin synthesis.

##### Targeting the PI3K/Akt/mTOR and Allied Cascades for Treating Vitiligo

In fact, many vitiligo treatments are taking advantage of activating the Akt/GSK-3ß/ß-catenin pathway [304]. PMPP, a synthetic chalcone derivative that targets MITF and tyrosinase, has been proposed for vitiligo treatment, as PMPP dose-dependently increases both tyrosinase and MITF activities. When PMPP, together with Akt inhibitor IV, was introduced to cells in vitro, tyrosinase activity was inhibited, resulting in a severe decrease in melanin content, suggesting that MITF needs the PI3K pathway and that PMPP is a promising agent for the treatment of vitiligo [301].

Zang et al. investigated D206008, a synthetic drug derived from the plant *Psoralencorylifolia* L., which is responsible for producing several psoralen derivatives that were earlier studied for pigmentation therapy. One of the most common psoralen derivatives studied is 8-methoxypsoralen (8-MOP), which indeed produces melanin effects, but it also has cytotoxic and gastrointestinal side effects [304]. D206008, a furocoumarin derivative, was tested for better pigmentation therapy, showing fewer side effects. D206008 was found to increase the melanogenesis (melanin) concentration via the activation of the Akt/GSK-3ß/ß-catenin pathway, which links to an increase in MITF and TYR activity, as well as melanogenesis, suggesting a possible vitiligo therapy [304].

Another potential therapy is Astragaloside IV (AS-IV), a natural flavonoid derived from the *Astragalus membranaceus* root with documented antioxidant, anti-inflammatory, anti-scar, and anti-hair loss effects. When compared to 8-MOP, AS-IV was found to have little to no cytotoxic effect but showed a dose-dependent increase in melanin synthesis, TYR activity, and a significant increase in p-ASKT and GSK-ß [304].

Hyperoside, from the Chinese medicine called Cuscutae Semen is another drug of interest for the treatment of vitiligo. It is a combination of the two roots *Cuscuta australis* and *Cuscata chinesis*. When exposed to melanocytes, hyperoside was found to increase melanocyte concentration and proliferation and protect against hydrogen peroxide-induced oxidation stress. One of these protective effects was found to occur through the PI3K/Akt pathway, as PI3K/Akt was increased upon the increase in hyperoside concentration. Activation of this pathway decreases the level of apoptosis, thus combating the premature apoptotic nature of hydrogen peroxide and increasing melanocyte cell survival [305].

*Gingko biloba* extract is a natural product that has commonly been used to treat vitiligo, due to its rich polyphenolic content that can provide antioxidant effects. In fact, studies have shown that the plant could decrease depigmentation and increase re-pigmentation in vitiligo patients. Egb761 is a standardized *Ginkgo biloba* extract known to have potent antioxidant activity and works by activating nuclear erythroid 2-related factor (Nrf2), which then activates the transcription of several downstream antioxidant genes such as the NQO-1, SOD2, and HO-1. Nrf2 is important for governing phase II detoxification and protects the cells from oxidative stress by activating the transcription of antioxidant genes [299,306].

## 5. Therapeutic Strategies for Chronic Immune-Mediated Inflammatory Skin Disorders

An array of agents from natural products, small molecules, and biologics are currently being employed to modulate the molecular pathological presentations of various skin inflammatory conditions. Natural products are effective therapeutic options owing to their ability to concurrently target multiple pathways that are involved in these disorders including the PI3K-AKT-mTOR, the core interest of the current review, most of which have low to no side effects even after long-term usage. Small molecules tend to be stable and are readily absorbed with multiple dosage forms available, though they most often have severe side effects, and long-term use may also cause drug tolerance. On the other hand, biologics are target-specific and present unequivocal advantages over other forms of molecules. Nevertheless, bioactive molecules often have limitations in their benefit owing to their unstable nature, complex structure, invasive parenteral route of administration, and/or skyrocketing costs. In general, the selection of a suitable molecule (small molecule, natural product, or biologics) for medicinal use depends on multiple factors, but most importantly on the benefit-to-risk ratio, availability, and cost.

### 5.1. Natural Products Targeting the PI3K/Akt/mTOR Pathway in Inflammatory Dermatoses

Natural products are substances derived from living organisms or simply organic compounds of biological origin, and exist as either primary or secondary metabolites, based on the intrinsic and extrinsic influences they pose for the survival of the organisms [306,307,308]. Primary metabolites (e.g., carbohydrates, lipids, and proteins) are generally indispensable for the growth, development, and reproduction of the organisms, whereas secondary metabolites (e.g., terpenoids, phenolic, and nitrogen/sulfur-containing compounds) are useful for ecological survival or defense [309].

Some active agents under the sub-classes of these main categories such as alkaloids, flavonoids, and tannins, have attained the emblem of dominance for their therapeutic worth over the years. In the past four decades, natural products-based drug discoveries have garnered considerable attention with an estimate of at least one-third of the top 20 newly approved drugs in the market originating from natural sources or were lead derivatives of natural products resources [310]. They have thus emerged as the best resource for drug discovery in lead identification as exemplified by the over 60% of currently marketed therapeutics entities and those at diverse stages of clinical development [310]. Researchers have attempted to explore the mechanism of action of various natural products on the modulation of PI3K-Akt-mTOR and associated signaling pathways as therapeutic targets for human diseases. The use of natural products for the management of chronic inflammatory skin disease has been in medical practice since before the inception of modern medicine [311]. Evolutional transition and natural selection producers of these agents account for their numerous pharmacological and biological activities that render them suitable for the management of these complex disease conditions. This adaptive modification is evident from the physiochemical and structural similarities existing between the members of various classes of natural products [312,313]. We discuss below various natural and related synthetic products that have been evaluated with promising potentials targeting the PI3K-Akt-mTOR and related pathways in diverse cutaneous inflammatory skin disorders.

#### 5.1.1. Flavonoids

Flavonoids are polyphenolic compounds, which are well-distributed in plants for their nutraceutical value and notable for their ability to sequester free radicals. In plants, flavonoids exist in free forms (aglycones) or in a glycosylated form, where a glycosidic bond is found in the A or C ring of the central core to produce O-glycoside or C-glycosides [314]. Chemically, flavonoids comprise about 12 classes, elaborating on a core phenyl-substituted benzopyran (e.g., 2-phenyl-4H-chromene, i.e., a C6-C3-C6 arrangement as the key scaffold with C6 being a benzene ring I, Figure 3) scaffold, and differing from each other prominently by the degree of oxidation of the C ring [11,13]. The resulting major classes include flavones, flavonols, flavans, flavanones, flavanonols, flavanols, chalcones and dihydrochalcones, isoflavones, aurones, anthocyanins, anthocyanidins, and catechins (Figure 3) [14]. These sub-families of secondary metabolites are utilized as prophylaxis or treatment options for various human diseases including cancer and especially chronic inflammatory disorders affecting the cardiovascular, nervous, muscular, respiratory, immune, digestive, integumentary, and other systems of the body [315,316]. This cocktail of benefits in relation to nutritional, medicinal and cosmeceutical value with no or minimal side-effects, affords them as appropriate alternative for the management of chronic conditions which mostly requires long-term usage of drugs [317].

A common long list of active flavonoid agents discovered in use or that are extensively being investigated against chronic inflammatory skin conditions include: apigenin, baicalin/baicalein, delphinidin, EGCG, fisetin, hersperidin, kaempferol, luteolin, naringenin, quercetin, tectogenin, wogonin, etc.

##### Apigenin

Apigenin (4′, 5, 7-trihydroxyflavone) is mostly found in plants used as vegetables, fruits, and herbs such as onion, chamomiles, spinach, orange, cilantro, parsley, sorghum, and rutabaga. It is an aglycone and exists abundantly in glycosylated form as either or C- glycosides with apiin, vitexin, isovitexin, rhoifolin, and schaftoside being the common native forms [318,319]. The medicinal value of apigenin spans from anti-inflammation to anti-diabetes, neuro-regeneration, anticancer, etc., [320]. Apigenin has been proven to effectively improve the dysfunctional epidermal permeability barrier that is often observed in inflammatory dermatoses [321,322], and has led to its modified dosage forms that are superior over or as substitutes for corticosteroids that are often used in inflammatory skin disorders [323].

##### Baicalin and Baicalein

Baicalin and baicalein, predominantly obtained from the root of *Scutellaria baicalensis* or Chinese skullcap in the family *Lamiaceae*, are flavones: the latter being the aglycone form of the former [324]. They are reported to exert anti-cancer, neuroprotective, cognitive enhancement, anti-ulcerative colitis, antioxidant, antiviral, photo-protective, and anti-inflammatory properties [325]. Baicalein has been shown to be therapeutically effective in reducing dermatophagoides pteronyssinus-induced atopic dermatitis-like skin lesions in NC/Nga mouse model [121]. It immunologically regulates innate immune toll-like receptors [326]. In psoriasis, the long-standing proof for the use of baicalin alternative medicine was established via in vivo studies, where common symptoms such as epidermal thickening, erythema, and desquamation backed with IL-17A, IL-22, and IL-23 level were significantly reduced after treatment [165].

##### Delphinidin

Delphinidin (3,3′,4′,5,5′,7-hexahydroxyflavylium) is a dominant dietary bioactive water-soluble anthocyanidin, abundantly found in pigmented fruits and vegetables including cranberries, bilberries, grapes, and pomegranates. Delphinidin and its glycosides possess pleiotropic biological effects including anti-inflammatory, antioxidant, antiproliferative, proapoptotic, anticancer, and prodifferentiation properties in diverse human diseases including inflammatory skin disorders [144,145,146,327]. Delphinidin has been shown to be beneficial in the treatment of psoriasis [140,144,145,146], and UVB-induced skin inflammation [147].

##### EGCG

EGCG (epigallocatechin-3-gallate) is the major catechin-based polyphenolic constituent of green tea with a limited amount in some fruits and nuts. Since ancient times, EGCG has been used in dietary adaptive interventions in the management of chronic conditions such as cardiovascular disease, diabetes, cancer, obesity, neurodegenerative disorders, stress, and inflammation [328,329,330]. Emerging findings from our and other research teams have identified the beneficial role of EGCG as well as the advantage of its nano-formulated form in reducing pathological biomarkers associated with psoriasis in preclinical psoriasiform models [127,331,332]. Mechanistically, EGCG, as a potential anti-itch agent, attenuates compound 48/80-induced itch [333] and imiquimod-induced chronic itch [334,335] behavior by suppressing the phosphorylation of ERK and Akt in the spinal cord of mice model. This was accompanied by a reduction in the expression of IL-23 mRNA, TRPV1 mRNA, and intracellular ROS in ND7-23 cells using chloroquine as an inducer [336]. It has been shown to have clinically improved acne in humans through the modulation of intracellular molecular targets as well as inhibiting P. acnes [337,338,339].

##### Fisetin

Fisetin (3,3′,4′,7-tetrahydroxyflavone) is a bioactive flavonol commonly found in pigmented fruits and vegetables such as strawberries, grapes, apples, persimmons, lotus root, and cucumber. Fisetin has antioxidant, antineoplastic, anti-inflammatory, senolytic, antiangiogenic, antidiabetic, antihyperlipidemic, antimicrobial, and cardio-protective effects [340,341,342,343,344]. Its effect on mTOR, NF-κB, lipoxygenase, glutathione, superoxide dismutase, and catalase, which are mostly involved in inflammation, provides general insight into the mechanism of action useful in the management of chronic inflammatory skin disease [345,346,347]. The therapeutic effect of fisetin has been observed in a chronic photo-damage in vivo *study*, where MMP-1, MMP-2, and COX-2 expressions were reduced along with restoration of the barricading characteristic of the skin in fisetin-treated mice [137]. Fisetin has also been shown to be beneficial in treating psoriasis [347] and AD in in vitro and in vivo models [348,349,350] with diverse molecular mechanisms.

##### Hesperidin

Hesperidin is a flavonone sourced mainly from citrus fruits and has pharmacological activity similar to the aforementioned polyphenolics, ranging from antihyperlipidemic, cardioprotective, antihypertensive, anticancer, antimicrobial to antidiabetic effects [351]. In the deglycosylated state, hesperidin is known as hesperidine, which has better bioavailability [352]. Its topical usage has been shown to improve the integrity of the epidermal permeability barrier function, which is mostly tampered with in aged skin and even most chronic skin inflammatory conditions [353]. In AD-like skin lesions in NC/Nga mice studies, hesperidin treatment reduced the levels of cytokines (IL-17 and IFN-α) levels, which are normally elevated in this disease [354].

##### Kaempferol

Kaempferol (3,5,7-trihydroxy-2-(4-hydroxyphenyl)-4H-1-benzopyran-4-one) is found in a variety of plants sources such as kale, cabbage, beans, and broccoli [355]. Its ability to downregulate ROS and cytokine involved in immune-mediated responses accounts for its therapeutic role in inflammatory and immune-mediated diseases [316]. Kaempferol has been demonstrated to effectively enhance the healing of chronic wounds. The healing of incisional and excisional wounds in both diabetic and non-diabetic rats was found to be significantly enhanced with a corresponding increase in wound tensile strength, closure, and reepithelization along with the levels of hydroxyproline and collagen in the wounds [356].

##### Luteolin

Luteolin (3′,4′,5,7-tetrahydroxyflavone) is a potent antioxidant and anti-inflammatory flavone obtained from a number of edible plants such as thyme, oregano, rosemary, celery, carrot, and peppermint [357]. Luteolin can inhibit or modulate pathways/molecules such as NF-κB, MAPK, PI3K as well as pro-inflammatory mediators, hence it has been used in the management of chronic inflammatory and metabolic diseases such as contact dermatitis, psoriasis, cancer, diabetes, neurocognitive and neurodegenerative disease [358,359,360].

##### Naringenin

Naringenin (5,7-dihydroxy-2-(4-hydroxyphenyl)-chroman-4-one) is a colorless and tasteless flavanone often found in citrus and tomato. This aglycone results from the cleavage of the sugar moiety in either of the following glycosides; naringin, narirutin, or naringenin-7-glycoside [361,362]. Its ability to induce an endogenous antioxidant effect in addition to the ROS scavenging potential of the hydroxyl groups makes it useful in the management of several metabolic and inflammatory conditions such as pain, atherosclerosis, obesity, diabetes, fibrosis, hepatosis, and cancer [363,364]. Recently it has also been demonstrated that naringenin has anti-inflammatory and anti-allergic features against arachidonic acid- (AA) and 12-O-tetradecanoylphorbol-13-acetate (TPA)-induced ear edema as well as IgE-mediated passive cutaneous anaphylaxis in vivo [365]. Furthermore, naringenin can significantly improve the construction of wound abrasions in vivo [366].

##### Quercetin

Quercetin (3,3′,4′5,7-pentahydroxyflavone) is an abundant pigment in plants such as apples, berries, citrus fruits, onion, broccoli, green tea and kale, and has anti-oxidant, anti-inflammatory, antiviral, antihypertensive, antihyperglycemic, psycho-stimulating and anticancer effects [367,368]. In skin, quercetin has been shown to mitigate the inflammatory responses in lipopolysaccharide (LPS)-induced RAW264.7 macrophages and 2,4-dinitrochlorobenzene (DNCB)-induced AD-like lesions mouse models. These responses were credited to its modulating the levels/activities of NF-κB, Erk1/2, JNK, IgE, and cytokines in chronic skin inflammatory conditions such as AD, psoriasis, and chronic wound [369,370,371].

##### Other Flavonoids

Other flavonoids, such as wogonin, nobiletin, and tectogenin, have also been identified for their salutary effects in chronic inflammatory skin conditions [372,373,374,375,376].

#### 5.1.2. Resveratrol and Analogues, Geniposide

Resveratrol (RSV), a natural phytoalexin abundantly present in grapes and berries, has been reported as a potent mTORC1 inhibitor and has the potential to treat chronic inflammatory cutaneous disorders [377,378]. It has been shown to be beneficial in enriching food including resveratrol-enriched rice-related clearance of skin inflammation and pruritus in mouse models of AD as well as the use of RSV in cutaneous function enhancement [379,380].

#### 5.1.3. Red Ginseng Extract

Red ginseng extract (RGE) is a popular traditional plant medicine prepared by repeatedly steaming and drying fresh ginseng to a moisture content of less than 15% to increase its bioactivity, which is frequently used in the East Asian countries including China, Japan, and Korea [101]. RGE is an anti-inflammatory agent with a demonstrated potential for AD treatment [101]. Osada-Oka et al. reported an anti-allergic effect of RGE through a significant reduction of 2,4-dinitrofluorobenzene (DNFB)-induced ear swelling and scratching behavior [101]. Treatment with RGE has been shown to decrease the chemokine C-C ligand 2 production in lesional keratinocytes in the AD mouse model [101].

### 5.2. Small Molecule Drugs

A majority of drugs in conventional medicine are small molecules, which are well-characterized organic or chemically synthesized active compounds with low molecular weights, making their pharmacokinetic and pharmacodynamic profiling straightforward [381]. Small molecules such as corticosteroids, immunomodulators, keratolytics, and antimicrobial drugs are often used to mitigate the discomforts associated with chronic inflammatory dermatoses. However, in order to minimize the side effects akin to these agents, topical formulations are mostly preferred.

#### 5.2.1. Corticosteroids

Corticosteroids are potent anti-inflammatory drugs that bind intracellularly to form steroid- receptor complex, which translocates into the nucleus to induce the production of annexin A1. This secondary messenger then inhibits the activation of phospholipase A2, which is necessary for the production of arachidonate-derived eicosanoids such as prostaglandin, prostacyclins, leukotrienes, and thromboxanes. Consequently, the inflammatory and mitogenic mediators that ignite the PI3K-Akt-mTOR, MAPKs, and other associated arms are precluded from being hyper-stimulated. Examples of commonly used steroids include clobetasol 17-propionate, betamethasone dipropionate, hydrocortisone, triamcinolone acetonide, fluocinonide, desonide and fluradrenolide [382].

#### 5.2.2. Immunomodulators

Immunomodulators are a cluster of agents that inhibit or activate the function of the immune system. They may act by inhibiting the activation/maturation of T-cells and reducing the number and mast cell degranulation, langerhans cells, inflammatory dendritic epidermal cells, the production of IL-2, 4 & 5, TNF-α, IFN-ɣ and the expression of the high-affinity receptor for immunoglobulin class E (FcεRI). In addition, some immunomodulators enhance the production of anti-inflammatory cytokines such as IFN-α, IL-1, 6, 8, 10, and 12. The macrolactams (tacrolimus, sirolimus, everolimus, and cyclosporine), contact sensitizers (diphenylcyclopropenone and dinitrochlorobenzene), immune-stimulators (imiquimod and resiquimod) and others (vitamin D3 analogs, anthralin, and zinc) are examples of immune-modulators with effects in cutaneous manifestations [159,167].

#### 5.2.3. Keratolytics

Keratolytics are topically formulated preparations, which soften and reduce the adhesion between keratinocytes. This thinning is achieved via the desquamation of hyper-keratotic epithelium and reduction in turnover of over-proliferative cells. Common keratolytics commercially available include anthralin, salicylic acid, retinoic acid, and coal tar [383,384].

#### 5.2.4. Antimicrobial Drugs

Antimicrobial drugs are pertinent for chronic inflammatory skin diseases (CISD) with bacteria or fungi as a causative or exacerbating factor, for instance, acne vulgaris (*Propionibacterium acnes*), AD (*Staphylococcus aureus* and *Malassezia* spp.) and buruli ulcer (*Mycobacterium ulcerans*). Potent agents with bacteriostatic, bactericidal, or fungicidal properties such as clindamycin, tetracyclines, rifampicin, ketoconazole, and fluconazole are employed [385,386].

### 5.3. Biologics

The future of protein-based therapeutics looks promising, considering the impact of biologics on numerous diseases. Biologics are products obtained using biotechnological tools from humans, animals, or microorganisms [387]. They include vaccines, gene therapies, blood components, whole blood, allergenics, recombinant fusion proteins, and many more that are also used in treating inflammatory skin disorders [388,389,390,391,392,393,394,395]. Under immune-mediated inflammatory disease, biologics are classified as, TNF inhibitors, interleukin (IL) inhibitors, B-cells inhibitors, and T-cells inhibitors (https://www.arthritis-health.com/treatment/medications/biologics-ra-and-other-autoimmune-conditions (accessed on 27 January 2023)).

#### 5.3.1. Tumor Necrosis Factor (TNF) Inhibitors

Tumor necrosis factor (TNF) inhibitors such as infliximab, adalimumab, and etanercept have been approved for the management of psoriasis and other inflammatory conditions [395], and have extensively discussed guidelines for such therapies [396]. They act by suppressing the activation of T-cells, macrophages, and B-cells, and the expression of IL-1, IL-6, IL-8, and CCL5, which are involved in inflammation [397]. Treatment with TNF inhibitors can effectively mitigate the influence of the PI3K-Akt on TNF and related signaling pathways, hence netting the signs and symptoms of CISD [398].

#### 5.3.2. Interleukin (IL) Inhibitors

Interleukin (IL) inhibitors are effective against CISD, owing to the involvement of IL family of cytokines in these conditions when dysregulated [399]. Ustekinumab, Secukinumab, Ixekizumab, Guselkumab, and Risankizumab are some examples, which act on targets known as IL proteins such as IL-1,12, 17, 23, and 24, hence decreasing the affinity to their natural receptors [400,401]. (https://go.drugbank.com/categories/DBCAT002105 (accessed on 24 January 2023)).

#### 5.3.3. B-Cells Inhibitors

B-cell inhibitors directly act against surface markers on B-cells or restrain the survival and signaling factors vital for their activation [402]. A typical example is ritumab, an anti-CD20 monoclonal antibody, which has been shown to be active in some inflammatory disorders including pemphigus, systemic sclerosis (SSs), AD, and Psoriasis [403].

#### 5.3.4. T-Cells Inhibitors

T-cell inhibitors have been applied in CISD, given the fact that T-cells play a crucial role in the advancement of CISD. Most of these inhibitors, such as etanercept, infliximab, adalimumab, uskekinumab, and siplizumab, act by halting the activation, proliferation, and migration of T-cells, thus alleviating the sign and symptoms of CISD [386,404,405]. 

Table 1, Table 2, Table 3 and Table 4 showcase a comprehensive compilation of available products, both natural and synthetic, employed in the management of various inflammatory skin conditions.

### 5.4. Antioxidant Natruceuticals

Guarneri et al. in their systematic review summarised several clinical studies that have investigated the potential of herbal derivatives and micronutrients in the treatment of inflammatory skin diseases. Curcumin, derived from turmeric, has shown efficacy in moderate to severe psoriasis, both topically and orally. Silymarin from milk thistle has demonstrated antioxidant properties and a reduction in depigmented areas in vitiligo patients. Anthocyanins, a group of polyphenols, have high bioavailability and have been effective in treating oral lichen planus (OLP). Ginkgo biloba extract, rich in polyphenols, has shown anti-inflammatory and antioxidant effects, resulting in repigmentation in vitiligo patients. Polypodium leucotomos, a tropical fern plant, exhibits antitumoral and anti-inflammatory properties, and when combined with narrowband ultraviolet B (NB-UVB), it promotes repigmentation in vitiligo. Purslane, an herbaceous weed, possesses anti-inflammatory, antioxidant, and immunoregulatory properties, leading to clinical improvement in OLP patients. Chamomile has active flavonoids with antioxidant and anti-inflammatory effects, providing relief in OLP symptoms [452,453,454].

## 6. Conclusions

The dysregulation of the PI3K/Akt/mTOR and associated signaling pathways is crucial in the development of chronic immune-mediated cutaneous disorders. Traditional immunomodulatory and therapeutic agents have shown limited efficacy against these diseases. However, phytochemicals have demonstrated promising results in targeting these signaling pathways and inhibiting skin cell proliferation and survival, with relatively low toxicities making them attractive adjuvant treatment options. Despite limitations in the availability of safe and durable remedies, lead compounds derived from natural and synthetic products hold the potential for managing chronic inflammatory skin diseases. Further investigations are necessary to test the efficacy of these strategies and assess the toxicity and adverse effects of lead therapeutic compounds on healthy tissues and systems. Nevertheless, the development of safe and efficacious lead compounds remains crucial in the fight against chronic inflammatory disorders. Future studies are necessary to identify bioactive compounds that can target molecular inflammation targets and decipher the underlying mechanisms of natural products and chemical libraries that can represent promising therapeutic leads. This review summarizes the progress made in synthetic small molecule inhibitors, biologics, phytochemicals, and extracts in developing therapies for chronic immune-mediated cutaneous disorders.

## Figures and Tables

**Figure 1 cells-12-01671-f001:**
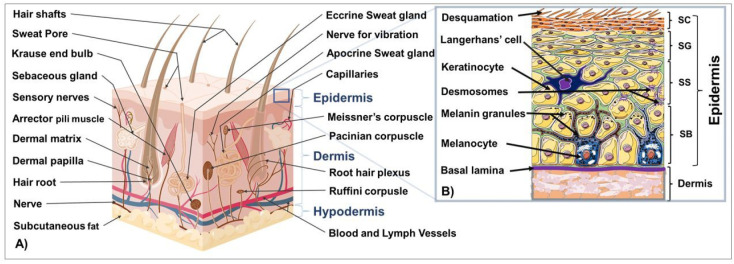
Schematic representation of the cross section of the skin (**A**) and structure of the epidermis and papillary dermis ((**B**), Inset).

**Figure 2 cells-12-01671-f002:**
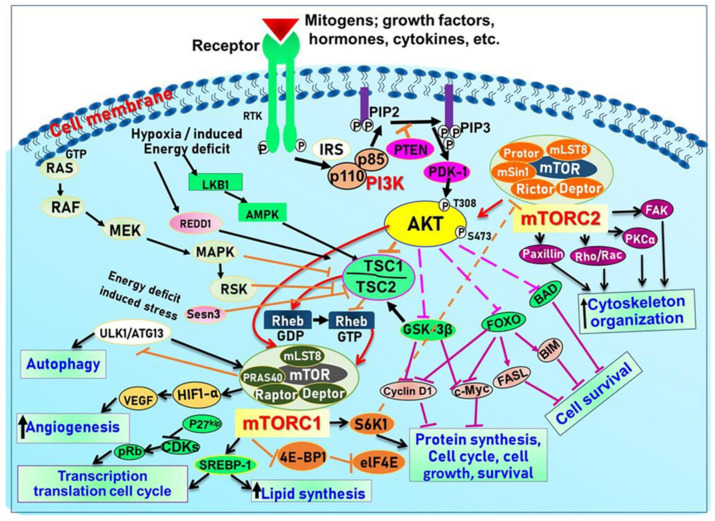
Schematic illustration of the PI3K/Akt/mTOR signaling pathway and the processes they regulate to control growth.

**Figure 3 cells-12-01671-f003:**
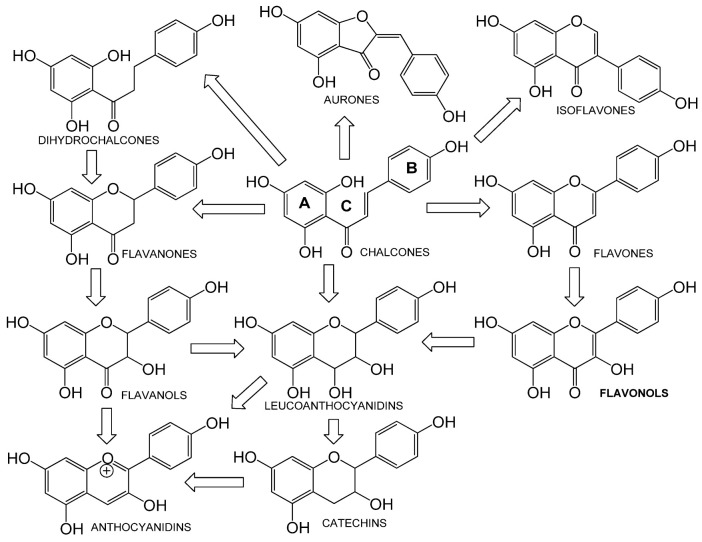
Core 2-phenyl-4H-chromene scaffold of flavonoids; different classes of flavonoids and their possible interchangeability are depicted.

**Table 1 cells-12-01671-t001:** Summary of Phytochemical Agents targeting the PI3K/Akt/mTOR Signaling Pathway for Chronic Inflammatory Skin Diseases Management.

Phytochemical	Class	Chemical Structures	Protein Target(s)	Chronic Skin Inflammation Type	References
Esculetin	Coumarin derivatives	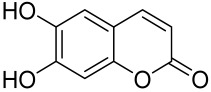	IKKα and P65	Psoriasis	[406]
ALA-PDT Aminolevulinic acid		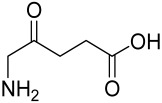	Akt-/Erk- mTOR -p70S6K pathway	Acne Vulgaris	[229]
Apigenin	Flavone	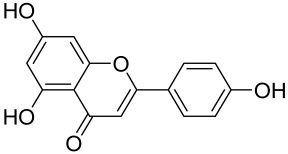		Inflammatory dermatoses	[318,319,320,321,322,323]
18-ß glycyrrhetinic acid	Pentacyclic triterpenoid	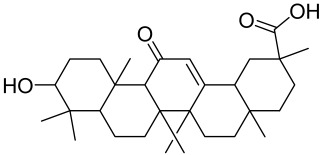	mTOR	Psoriasis	[151]
Caffeine	Methylxanthine	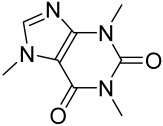	mTOR	Mycosis fungoides, pemphigus vulgaris, alopecia areata	[96]
Curcumin	Diarylheptanoid	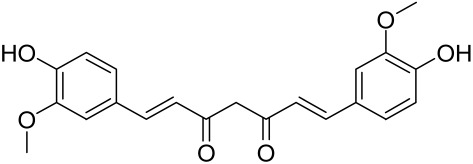	mTOR/Akt/S6K1	Mycosis fungoides, pemphigus vulgaris, psoriasis	[96]
Epigallocaechin-3- gallate	Catechin	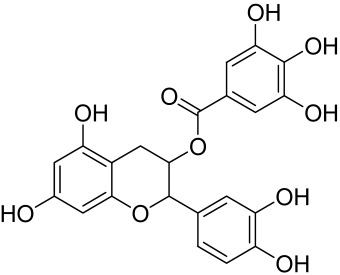	mTOR/Akt	Mycosis fungoides Pemphigus vulgaris Acne vulgaris	[330,331,332]
Erianin	Bibenzyl	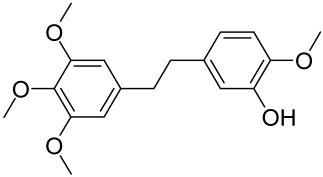	Akt/mTOR pathway	Psoriasis	[154]
Gartanin	Xanthones	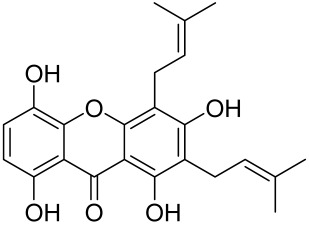	mTOR/Akt/S6K1	Mycosis fungoides, pemphigus vulgaris, psoriasis	[96]
Hesperidin	Bioflavonoid	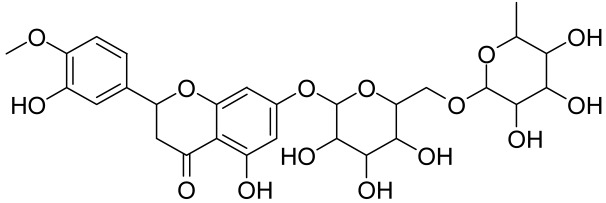	IL-17 and IFN-α axis	Atopic dermatitis	[352,353,354]
Isotretinoin/Retionoic acid	Retinoid	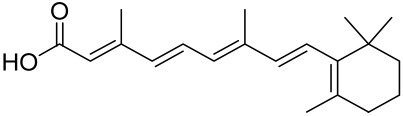	FoxO1 mTORC1	Acne Vulgaris, hidradenitis suppurativa	[244,259]
Imperatorin	Furocoumarin	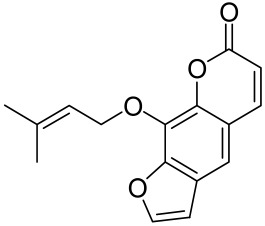	IGF-1/Akt PPAR-y	Acne Vulgaris	[304]
Kaempferol	Flavonol	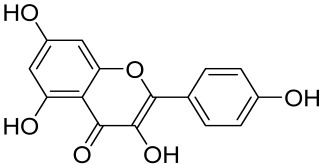	Reactive oxygen species pathway	Chronic wounds	[355,356]
Luteolin	Flavone	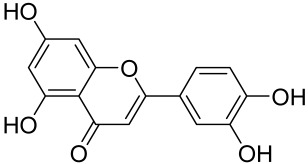	NF-κB, MAPK, PI3K	Contact dermatitis Psoriasis	[357,358,359,360]
Naringenin	Flavanones	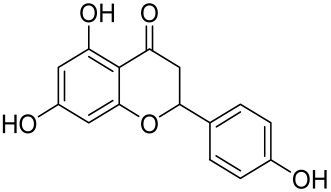	Reactive oxygen species pathway	Wounds	[361,362,363,364,365,366]
Matrine	Alkaloid	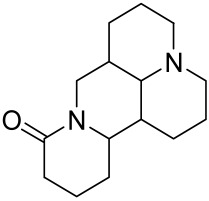	PI3K/Akt/mTOR pathway	Psoriasis	[153]
Paeoniflorin	Monoterpene glycoside	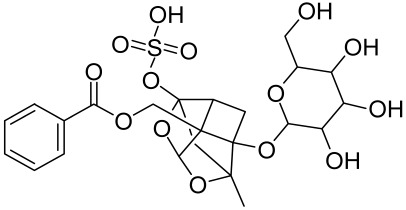	PI3K/Akt/mTOR pathway	Psoriasis	[152]
Psoralen derivatives	Linear furanocoumarins	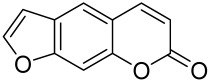	Akt/GSK-3ß/ß-catenin pathway	Vitiligo	[304]
Resveratrol	Stilbenol	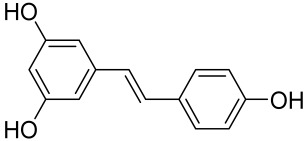	PI3K/mTOR	Mycosis fungoides Pemphigus vulgaris Acne vulgaris	[96,407]
Quercetin	Flavonoid	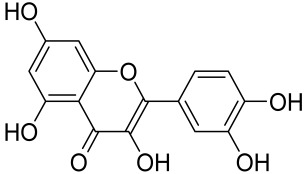	Akt	Mycosis fungoides, pemphigus vulgaris, acne vulgaris	[96]
Obacunone	Tetranortriterpenoids	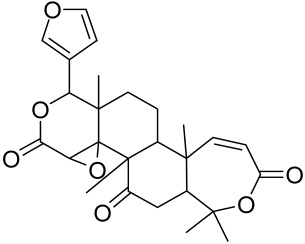	Akt	Atopic dermatitis	[408]
Obaculactone		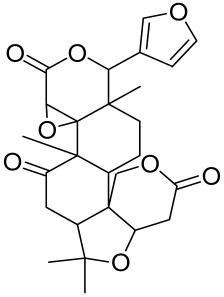	Akt	Atopic dermatitis	[408]
Ginsenoside Rb1	Triterpenoid saponins	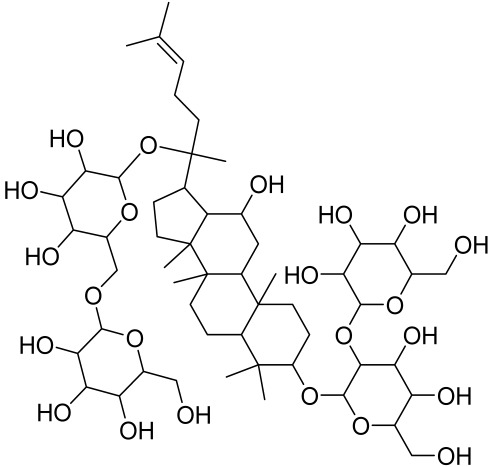	Akt	Alopeciea areata	[409]
Ginsenoside Rd	Triterpenoid saponins	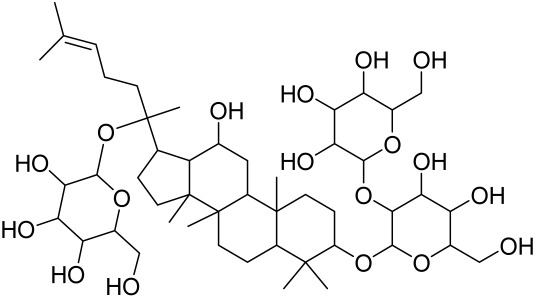	Akt	Alopeciea areata	[409]
Ginsenoside Rh3	Triterpenoid saponins	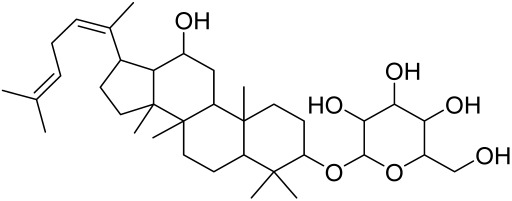	Akt	Alopeciea areata	[409]
Ginsenoside Rg1	Triterpenoid saponins	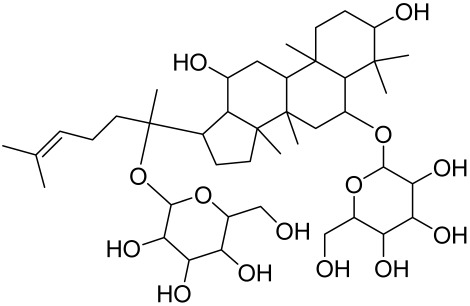	Akt	Alopeciea areata	[409]
Ginsenoside Rg3	Triterpenoid saponins	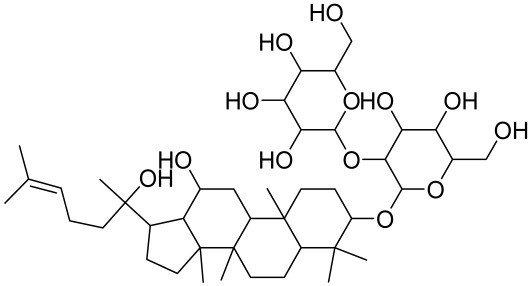	Akt	Alopeciea areata	[409]
Rapamycin	Macrolide	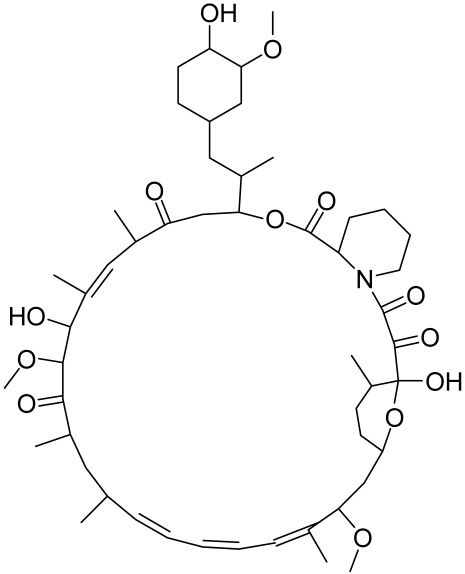	mTOR	Psoriasis Systemic sclerosis	[408,409,410,411]
Wortmannin	Oxa-steroid.	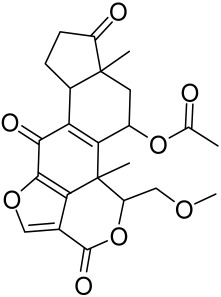	PI3K/Akt/ mTOR pathway	Skin atrophy	[412]
α-Mangostin	Xanthones	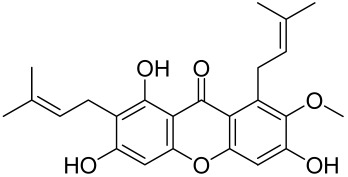	MAPK downstream proteins	Antiaging, anti-wrinkle	[413]
Rhododendrin	Arylbutanoid glycoside	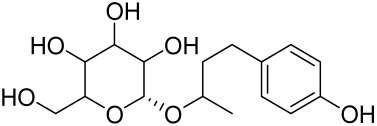	PI3K/Akt/mTOR pathway	Psoriasis	[414]
3β,6β,16β-Trihydroxylup-20(29)-ene	Pentacyclic triterpene	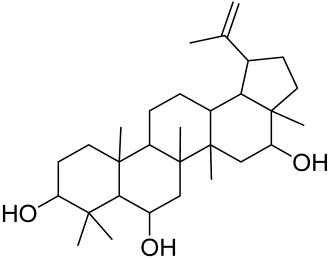		Psoriasis	[415]
Geniposide	Iridoid glycoside	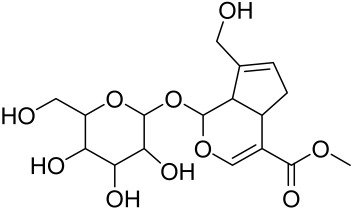	Akt/mTOR	Systemic sclerosis	[416]
Delphinidin	Anthocyanidin	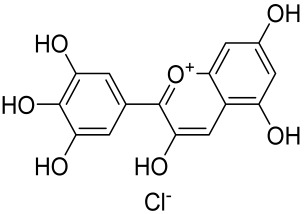	PI3K, mTOR/p70S6K	Psoriasis	[140]
Fisetin	Flavonol	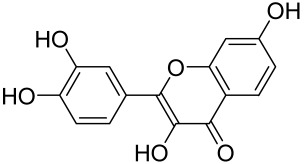	Akt/mTOR, p70S6K	Psoriasis, UV-induced skin inflammation	[417,418,419]

**Table 2 cells-12-01671-t002:** Summary of extracts used for Chronic Inflammatory Skin Diseases Management.

Phytochemical	Class	Protein Target(s)	Chronic Skin Inflammation Type	References
*Actinidia arguta*	Extract	mTOR	Atopic Dermatitis	[110]
Crude extract of Carinianadomestica fruit peels (CdE)	Extract	-	Acute and chronic irritant contact dermatitis	[419]
Tabernaemontanacatharinensis leaf extract (snakeskin)	Extract	-	Irritant contact dermatitis, skin edema	[420]
Nasturtium officinale (watercress)	Extract	-	Irritant contact dermatitis	[421]
PSORI-CM02	Herbs mixture	PI3K/Akt/mTOR	Psoriasis	[422]
Honey	Polyphenols and other antioxidants	-	Wounds	[423]

**Table 3 cells-12-01671-t003:** Summary of Synthetic Agents used In Chronic Inflammatory Skin Diseases.

Active Agent	Class	Chemical Structures	Protein Target(s)	Chronic Skin Inflammation Type	References
Everolimus	Immunomodulator/macrolactams	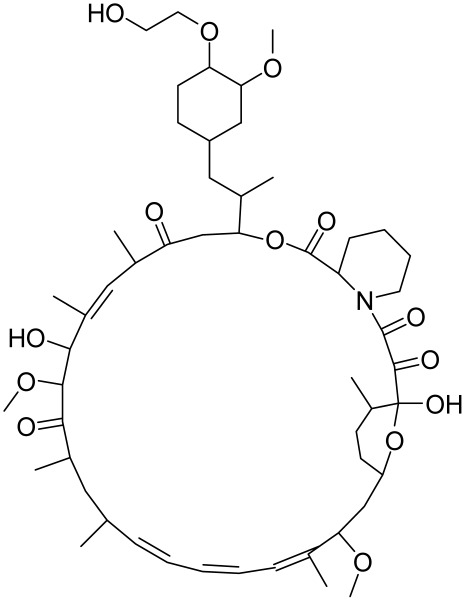	mTOR	Psoriasis and tuberous sclerosis	[400,401]
Anthralin	Immunomodulator	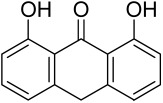	IL, TNF-α and IFN-α axis	Alopecia areata and psoriasis	[159,167,424]
Betamethasone dipropionate	Corticosteroid	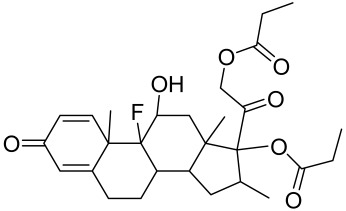	IL and IFN-α axis	Psoriasis and dermatitis	[382]
BEZ235	Kinase inhibitor	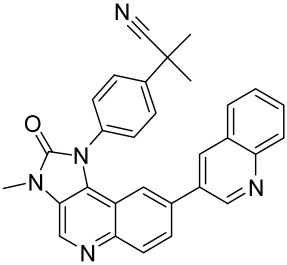	PI3K/Akt/mTOR pathway	Systemic sclerosis	[170]
Cyclosporine	Immunomodulator/macrolactams	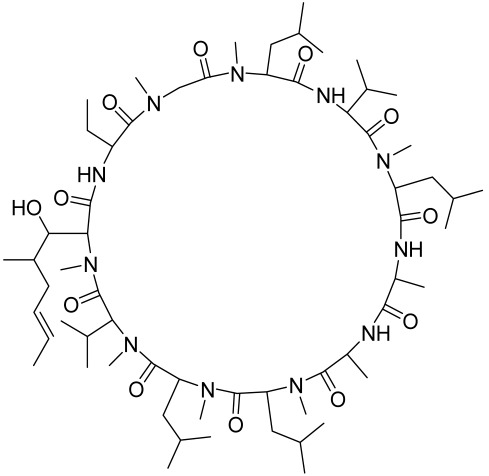	IL and IFN-α axis	Atopic dermatitis and psoriasis	[425]
Clobetasol 17-propionate	Corticosteroid	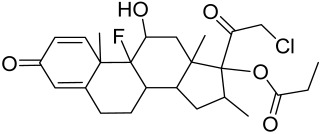	IL and IFN-α axis	Psoriasis and dermatitis	[382,426,427]
Coal tar	Keratolytics		Aryl hydrocarbon receptor	Atopic dermatitis and psoriasis	[428]
Desonide	Corticosteroid	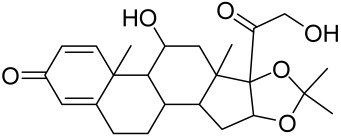	IL and IFN-α axis	Atopic dermatitis and psoriasis	[382,428,429,430]
Dinitrochlorobenzene	Immunomodulator/contact sensitizers	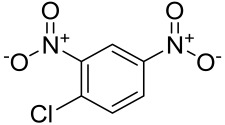	IFN-γ, IL-1β, and IL-2,	Alopecia areata	[431]
1α,25-dihydroxy vitamin D3	Vitamin D analog	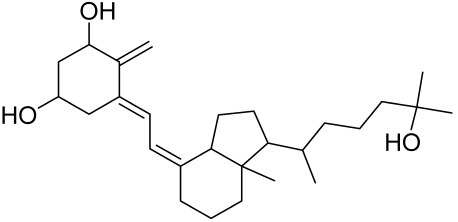	Akt/mTOR	Psoriasis	[432,433]
Diphenylcyclo-propenone	Immunomodulator/contact sensitizers	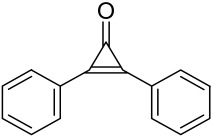	IFN-γ, IL-1β, and IL-2,	Alopecia areata	[431]
Sirulumus	Immunomodulator/macrolactams	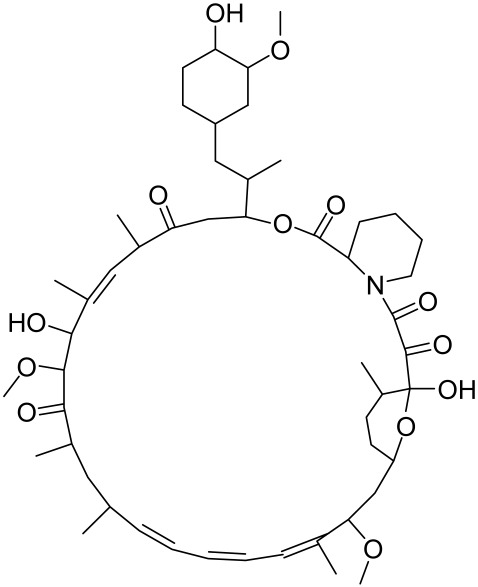	mTOR	Psoriasis and tuberous sclerosis	[400,401]
Fluocinonide	Corticosteroid	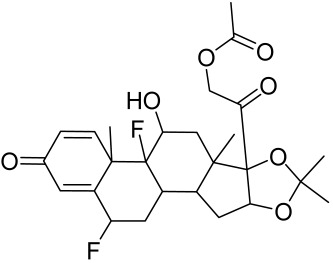	IL and IFN-α axis	Oral lichen planus and alopecia areata	[96,97,434]
Tacrolimus	Immunomodulator/macrolactams	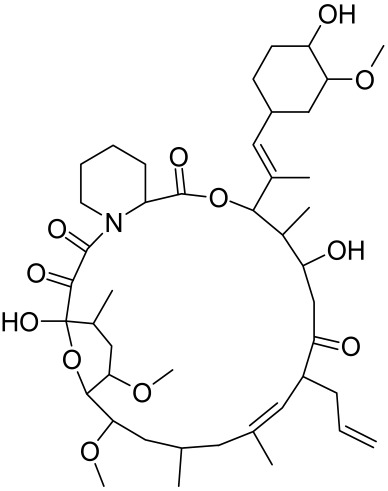	mTOR	Psoriasis and tuberous sclerosis	[400,401]
Erlotinib	Kinase inhibitor	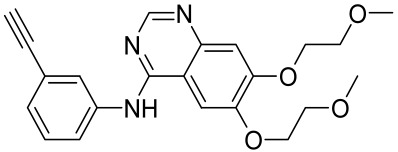	PI3K/Akt/ mTOR pathway	Pemphigus vulgaris	
Hydrocortisone	Corticosteroid	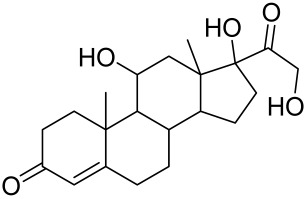	IL and IFN-α axis	Atopic dermatitis and psoriasis	[98,435,436]
Ketoconazole	Antimicrobial	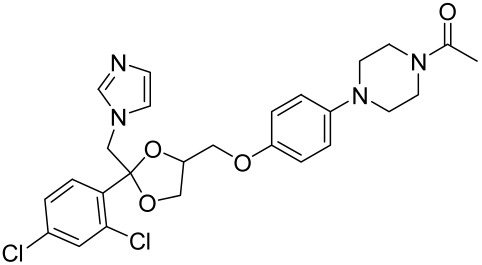		Acne vulgaris	[99,437]
Metformin	Biguanides	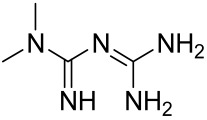	mTOR	Diabetic foot ulcer	[403,438]
PP1{4-amino-5-(4-methylphenyl)-7-(t-butyl)pyrazolo[3,4-d]- pyrimidine}	Kinase inhibitor	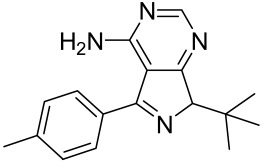	PI3K/Akt/ mTOR pathway	Pemphigus vulgaris	[439]
Rifampicin	Antimicrobial	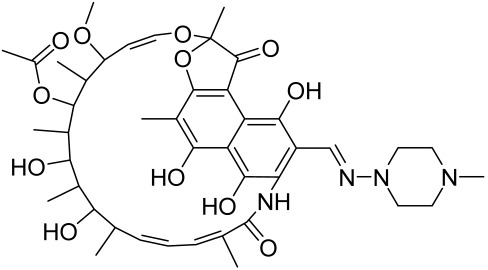		Buruli ulcer	[243]
Salicylic acid	Keratolytics	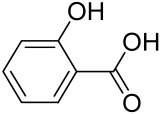		Acne vulgaris and psoriasis	[439,440]
Triamcinolone acetonide	Corticosteroid	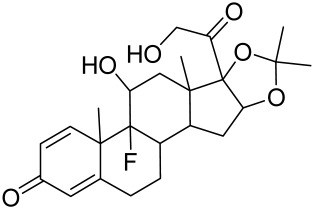	IL and IFN-α axis	Acne vulgaris, hidradenitis suppurativa and psoriasis	[259,441,442]
Zinc	Immunomodulator		IL axis and Reactive oxygen species (ROS)	Acne vulgaris, vitiligo	[443,444,445]

**Table 4 cells-12-01671-t004:** Summary of Biologics used In Chronic Inflammatory Skin Diseases.

Active Agent	Class	Protein Target(s)	Chronic Skin Inflammation Type	References
Secukinumab	IL inhibitors	IL axis	Psoriasis	[395,446,447]
Adalimumab	TNF and T-cells inhibitor	TNF axis	Psoriasis and hidradenitis suppurativa	[448]
Etanercept	TNF and T-cells inhibitor	TNF axis	Psoriasis	[419]
Dupilumab	IL inhibitors	IL axis	Atopic Dermatitis	[126]
Guselkumab	IL inhibitors	IL axis	Psoriasis	[449]
Infiximab	TNF and T-cells inhibitor	TNF axis	Psoriasis and hidradenitis suppurativa/acne inversa	[259]
Ixekizumab	IL inhibitors	IL axis	Psoriasis	[394,450]
Risankizumab	IL inhibitors	IL axis	Psoriasis	[450,451]
Ustekinumab	IL inhibitors	IL axis	Psoriasis	[126,448]

## Data Availability

Not applicable.

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
