# Peer review of "The PI3K-Akt-mTOR and Associated Signaling Pathways as Molecular Drivers of Immune-Mediated Inflammatory Skin Diseases: Update on Therapeutic Strategy Using Natural and Synthetic Compounds"

_cells, 2023, doi:10.3390/cells12121671_

Round 1
Reviewer 1 Report
This review is very interesting and appropriately cover the field indicated in the Title.
Minor issues: Some flavonoids have been demonstrated to be active also on other inflammation patways important in Psoriasis and AD. If possible please add this concept in the paragraph
Author Response
Reviewers Comments to Author:
Reviewer #1:
Concern 1: Some flavonoids have been demonstrated to be active also on other inflammation patways important in Psoriasis and AD. If possible please add this concept in the paragraph.
Response: Thank you for your insightful comment. Based on your suggestion, we have addressed the additional pathways influenced by flavonoids. Specifically, we have included discussions on the suppression of MMP-1, MMP-2, COX2, and cytokine inhibition mediated by flavonoids. Please see page 17, lines 489-517. We believe that these additions will enhance the comprehensiveness of our study and further highlight the multifaceted effects of flavonoids on cellular disease processes.
Reviewer 2 Report
The paper is a huge analysis regarding the PI3K/Akt/mTOR axis in skin immune-related disease.
The first part of the paper analyse skin and its structure together with risk factors for developing inflammation and the relative pathways. The PI3K/Akt/mTOR axis plays a critical role in regulating multiple physiological processes, such as cell growth, proliferation, autophagy, apoptosis, migration, invasion, differentiation, metabolism, and angiogenesis.
Figure 2 explains the complex mechanism of the axis.
Subsequently the authors successfully span several immune-related disease and efficently focus on the role and the potential application of this axis.
The manuscript is the result of a big effort by the researchers. Accorging to my opinion it could be improved by better defining epigenetic modification in skin disorders and nutraceuticals role in cutis inflammation by evaluating references such as 10.2174/1566524020666200812222324 and https://doi.org/10.3390/antiox10071087.
Author Response
Reviewer #2:
Concern 1: Accorging to my opinion it could be improved by better defining epigenetic modification in skin disorders and nutraceuticals role in cutis inflammation by evaluating references such as 10.2174/1566524020666200812222324 and https://doi.org/10.3390/antiox10071087.
Response: Thank this reviewer for his/her valuable suggestion. We have made significant revisions to the manuscript to include discussions on the epigenetic risk factors involved in the onset of inflammatory skin diseases. Please see Page 3 lines 99-1117 for more information. Additionally, we have included a new sub-heading discussing the role of antioxidant nutraceuticals in the management of inflammatory skin diseases (see page 56 lines 2099-2117).
Reviewer 3 Report
1) This manuscript is too diverse and does not focus on PI3K-Akt-mTOR, so it is recommended to delete interleukin stories such as ROS, TNF-alpha, IFN-gamma, and IL-17, 22, & 23 etc.
2) Similarly in Table 1 & Table 2, phytochemicals unrelated to mTOR can be deleted.
3) In line 90, there are many environmental factors as risk factors for exacerbation and onset of inflammatory skin diseases. Therefore, risk factors for inflammatory skin diseases should include humidity, temperature, and air pollutants.
4) Abbreviations need to be re-organized.
a. In line 123, 263,280, 303, 693, 1254, 1295, 1346, and 1409, Atopic dermatitis ïƒ AD
b. In line 1433, interleukinïƒ IL
Author Response
Reviewer #3:
Concern 1: This manuscript is too diverse and does not focus on PI3K-Akt-mTOR, so it is recommended to delete interleukin stories such as ROS, TNF-alpha, IFN-gamma, and IL-17, 22, & 23 etc.
Response: We appreciate the very informative feedback from this reviewer regarding the focus of our manuscript on the PI3K/AKT/mTOR pathway. We have carefully considered this suggestion and have revised the manuscript to remove redundancies and ensure a more streamlined focus on the PI3K/AKT/mTOR pathway. However, as requested by other reviewers, cytokines such as TNF-alpha, IFN-gamma, and IL-17, 22, and 23 also impact and modulate the PI3K/AKT/mTOR axis. While we have reduced the discussion to maintain a clearer focus, we briefly addressed the involvement of these cytokines, and believe it would enhance the completeness of the current manuscript version. These brief mentions will contribute to a better understanding of the broader effects and implications of the PI3K/AKT/mTOR pathway. We appreciate the opportunity to clarify this and believe that incorporating these aspects will improve the overall scientific value of the manuscript.
Concern 2: Similarly in Table 1 & Table 2, phytochemicals unrelated to mTOR can be deleted.
Response: We have carefully addressed these concerns and suggestion and have revised the manuscript to remove redundancies to now ensure a more streamlined focus on the PI3K/AKT/mTOR pathway in the tables.
Concern 3: In line 90, there are many environmental factors as risk factors for exacerbation and onset of inflammatory skin diseases. Therefore, risk factors for inflammatory skin diseases should include humidity, temperature, and air pollutants.
Response: We thank the reviewer for his/her constructive and insightful comments and suggestions. We have now revised the paragraph based on your suggestions and have included those recommended risk factors. Please refer to Page 3, lines 99-117.
Concern 3: Abbreviations need to be re-organized. a. In line 123, 263,280, 303, 693, 1254, 1295, 1346, and 1409, Atopic dermatitis ïƒ AD. b. In line 1433, interleukinïƒ IL
Response: We thank this reviewer for his/her valuable feedback on our manuscript. We have made the requested and necessary corrections and revisions to improve the clarity and accuracy of the manuscript. We have equally corrected the abbreviations as suggested. Thank you so very much for your time and feedback